# Code Graph Model (CGM):
# A Graph-Integrated Large Language Model for Repository-Level Software Engineering Tasks

**Hongyuan Tao**[1][*] **Ying Zhang**[12][*] **Zhenhao Tang**[1][*] **Hongen Peng**[1]  **Xukun Zhu**[13]  **Bingchang Liu**[1]

**Yingguang Yang**[1]  **Ziyin Zhang**[14]  **Zhaogui Xu**[1]  **Haipeng Zhang**[2]  **Linchao Zhu**[3]  **Rui Wang**[4]

**Hang Yu**[1][†] **Jianguo Li**[1][†] **Peng Di**[1][†]

[1]Ant Group,  [2]ShanghaiTech University,  [3]Zhejiang University,  [4]Shanghai Jiaotong University

{hyu.hugo,lijg.zero,dipeng.dp}@antgroup.com

## Abstract

Recent advances in Large Language Models (LLMs) have shown promise in function-level code generation, yet repository-level software engineering tasks remain challenging. Current solutions predominantly rely on proprietary LLM agents, which introduce unpredictability and limit accessibility, raising concerns about data privacy and model customization. This paper investigates whether open-source LLMs can effectively address repository-level tasks without requiring agent-based approaches. We demonstrate this is possible by enabling LLMs to comprehend functions and files within codebases through their semantic information and structural dependencies. To this end, we introduce Code Graph Models (CGMs), which integrate repository code graph structures into the LLM's attention mechanism and map node attributes to the LLM's input space using a specialized adapter. When combined with an agentless graph RAG framework, our approach achieves a 43.00% resolution rate on the SWE-bench Lite benchmark using the open-source Qwen2.5-72B model. This performance ranks first among open weight models, second among methods with open-source systems, and eighth overall, surpassing the previous best open-source model-based method by 12.33%.[3].

## 1 Introduction

The dream of automating software engineering (SE) has long captivated both the SE and artificial intelligence (AI) communities [1, 2, 3]. Recent advancements in Large Language Models (LLMs) have shown promising results, particularly in code generation at the function level, with models achieving resolution rates above 90% on benchmarks such as HumanEval [4]. Unfortunately, real-world SE tasks extend far beyond isolated functions or self-contained code files. This is exemplified by repository-level issue resolution [5, 6], which encompasses not only software maintenance—addressing bugs and technical debt—but also software evolution, which involves introducing new features and enhancements [7].

---

[*]Equal contribution.

[†]Corresponding authors.

[3]The code is available at https://github.com/codefuse-ai/CodeFuse-CGM

The complexity of repository-level coding tasks has led researchers and practitioners to assume that sophisticated strategies are necessary for their completion [8]. Indeed, current leading approaches typically utilize **LLM agents powered by proprietary models** like GPT-4/4o [9] and Claude 3.5 Sonnet [10]. These agents are designed to leverage tools, execute commands, observe environmental feedback, and plan subsequent actions [11]. Nevertheless, these methods suffer from two problems. First, **the agent-driven mechanism** introduces unpredictability in decision-making [2]. As the reasoning processes become intricate in tackling complex problems, the accumulation of errors can hinder the generation of optimal solutions [12]. Second, **the reliance on closed-source models** creates substantial barriers for the broader SE community [13, 14], including limited accessibility, inability to enhance or customize models for specific tasks, and serious security concerns regarding the privacy of sensitive code repositories when interacting with external API services.

The above two challenges lead to a bold question: Can **open-source LLMs** be employed in an **agentless manner** to complete repository-level coding tasks? At first glance, this seems improbable. Closed-source agent-based approaches can resolve up to 55% of issues on the popular SWE-bench Lite benchmark[4] for issue fixing, whereas existing methods using open-source models have only achieved a maximum resolution rate of 30.67% as of May 2025 [15].

Despite these initial reservations, we posit that the answer is "Yes", and the key lies in **empowering the open-source LLMs to fully comprehend code repositories, not just the information within individual functions and files, but also the dependencies across functions and files.** To move forward to this goal, we propose **Code Graph Models (CGMs)**, to jointly model the semantic and structural information of code repositories. Specifically, we first construct a code graph for each repository, which characterizes the hierarchical and reference dependencies between code entities. We then develop a method to integrate this graph into the LLM through two key mechanisms. (i) **Semantic Integration**: **Node attributes (containing code or comments)** are first encoded by a pretrained text encoder and then mapped to the LLM's input space via an adapter, enabling the model to understand the semantic information of all nodes. (ii) **Structural Integration**: The **graph structure** is incorporated into the LLM through the attention mask, allowing direct message passing only between neighboring nodes in each layer of the LLM, similar to spatial Graph Neural Networks (GNNs) [16]. The entire system—comprising the text encoder, adapter, and LLM decoder—is then fine-tuned using Low Rank Adaptation (LoRA) [17]. The resulting CGM can tackle repository-level coding tasks by using both the code graph and user instructions (text format). To further augment the abilities of the CGM, we develop a specially designed Graph Retrieval-Augmented Generation (RAG) framework, consisting of four modules: Rewriter, Retriever, Reranker, and Reader (i.e., CGM). The first three modules focus the CGM on the subgraph that is most pertinent to the user's query or issue.

Our approach has demonstrated remarkable results on the SWE-bench Lite benchmark, **reaching a 43.00% resolution rate using the open-source Qwen2.5-72B model and our agentless RAG framework**. As of May 2025, this performance ranks first among methods utilizing open-source models, second among methods with open-source code implementations (the underlying model may still be closed-source), and eighth overall. Notably, our approach surpasses the previous best method based on open-source models (Moatless+DeepSeek-V3 [15]) by **12.33%**, despite that method employing DeepSeek-V3, which shows stronger performance than Qwen2.5-72B.

The main contributions of this work are as follows:
- We propose CGMs, a novel architecture that seamlessly integrates repository code graphs with open-source LLMs through semantic and structural integration.
- We develop an agentless Graph RAG framework that enhances the CGM's performance by focusing on the most relevant subgraphs for user queries.
- Our CGM, armed with the Graph RAG, achieves a 43.00% resolution rate on SWE-bench Lite, surpassing most agent-based approaches. We also demonstrate its effectiveness on other repository-level tasks such as code completion.

## 2   Related Work

**Large Language Models for Code**   Recent advancements in LLMs have shown remarkable success in generating code at self-contained function or file levels [3]. This includes powerful closed-source models like GPT-4/4o [9], Gemini-2.0 [18], and Claude 3.5 Sonnet [10], as well as open-source alternatives such as Llama 3.1 [19], Qwen 2.5 [20], and DeepSeek-V3 [21]. Additionally,

---

[4]https://www.swebench.com/

code-specialized open-source models have also emerged, including CodeFuse [22, 23, 24], Code Llama [25], StarCoder [26, 27], DeepSeek-Coder [14, 28], and Qwen-Coder [29]. However, these models struggle with repository-level coding tasks that better reflect practical software development scenarios. Even the most capable closed-source models achieve only modest success rates on the SWE-bench Lite benchmark [5] for real-world issue fixing, while open-source models lag further behind with a maximum resolution rate of 26% [30]. Although closed-source models show superior performance, their limited accessibility and data privacy concerns hinder widespread adoption in the SE community. Furthermore, their proprietary nature prevents fine-tuning on task-specific data to improve performance, if even such data is available.

For open-source LLMs to better handle repository-level tasks, they must develop a comprehensive understanding of both semantic and structural information within codebases. DeepSeek-Coder [14] has attempted to address this challenge by pre-training models on topologically sorted repository codes. However, this approach faces two major limitations: real-world repositories often contain more code than can fit within the model's maximum context length; and the conversion of repository structure into text format tends to obscure explicit dependencies that exist in the codebase.

To overcome these challenges, we propose representing repositories as text-rich graphs and aligning them with LLMs via self-supervised continual pre-training. This approach preserves code repository structure while enabling more effective processing and understanding of complex dependencies.

**Graphs in Code Language Models**    The integration of graph structures into code language models can be classified into three main approaches [31]: (1) attention mask modification, (2) graph-to-text conversion, and (3) positional encoding augmentation. In the first approach, models like GraphCode-BERT [32] and StructCoder [33] modify attention masks to capture relationships between code tokens in Abstract Syntax Trees (ASTs) and Data Flow Graphs (DFGs). The second approach, demonstrated by TreeBERT [34] and UniXcoder [35], transforms ASTs or node paths into textual sequences that can be processed by language models. The third approach, exemplified by TPTrans [36], leverages relative positional encodings to represent structural relationships within ASTs.

While these approaches have shown promise, they primarily focus on Transformer encoders and small-scale language models (such as BERT or CodeT5) and are limited to file- or function-level tasks. In contrast, our work enhances decoder-only LLMs to handle repository-level tasks. We construct text-rich code graphs for entire codebases, moving beyond simple ASTs or DFGs. Inspired by GraphCodeBERT and StructCoder, we incorporate graph structures through attention masks in LLMs. However, due to the text-rich nature of the graphs, each node's text or semantic information is processed by a pretrained text encoder and then projected onto the LLM's input space via an adapter.

**Agent-drive Methods for Software Engineering**    LLM-based agents like Devin [37] have shown the potential to solve real-world SE problems through their reasoning [38, 39] and interactive capabilities [40, 41, 42, 11]. Along this direction, researchers have worked to enhance LLM agents through various approaches, including specialized agent-computer interfaces (ACI) [43, 44, 8, 45], fine-grained search [46, 12, 11], and expanded action spaces [47].

However, these agent-based approaches face several drawbacks. First, they typically delegate decision-making to the agents, allowing them to determine both the timing and nature of actions. While agents base their decisions on previous actions and environmental feedback, the expansive action space and complex feedback mechanisms can lead to repetitive behaviors or accumulating errors, ultimately resulting in suboptimal solutions [12]. Second, resolving a single issue often requires 30-40 interaction turns, making the process time-consuming and complicating the identification of specific turns that resulted in unsatisfactory outcomes [2]. Third, the inherent unpredictability of agent behavior and reliance on closed-source models creates obstacles for leveraging data to improve performance, despite the abundance of such data in practice, such as issue-patch pairs for issue fixing [5]. While SWE-Gym [48] attempts to address trainability, it may introduce bias by only training with the trajectories that lead the SWE agent to correct answers. As a remedy, we propose the CGM, built on open-source LLMs and enhanced through an agentless Graph RAG framework.

**Agentless Methods for Software Engineering**    Agentless models offer a more controlled approach to simulating real-world SE processes by following well-defined, fixed steps rather than relying on LLM agents to make autonomous decisions or use complex tools. They help avoid the issues of unpredictability and lengthy interaction chains. These methods typically operate in two main stages: localization and editing [49]. The localization stage identifies relevant code snippets within a repository, while the editing stage generates or modifies code based on these identified sections.

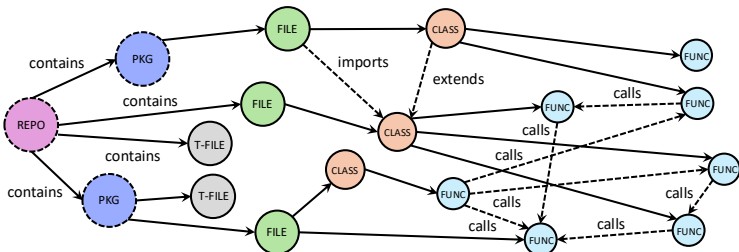

Figure 1: An example of our repository-level code graph. "PKG", "FUNC", and "T-FILE" represent "PACKAGE", "FUNCTION", and "TEXTFILE", respectively. Solid and dashed lines indicate hierarchical (contains) and reference dependencies (calls/imports/extends), respectively.

This framework is particularly effective for repository-level code completion tasks, especially when combined with RAG [50, 51]. For more complex tasks like issue fixing, enhanced approaches with additional steps exist [2, 1]. For instance, Agentless [2] implements a comprehensive ten-step pipeline, dedicating four steps to improving localization accuracy. This method has achieved a promising resolution rate of 40.67% on SWE-bench Lite, comparable to state-of-the-art (SOTA) agent-based methods, though it relies on the closed-source model Claude-3.5 Sonnet.

Recent research has also focused on enhancing code understanding by incorporating structural information through graph-enhanced repository modeling [52, 53, 49]. However, even when graph structures are used during retrieval, existing methods typically flatten the retrieved code snippets into linear text sequences for downstream model prompting. This flattening process fails to preserve the inherent heterogeneity between graph and text modalities. As a remedy, we propose the CGM that explicitly aligns these two distinct modalities, enabling better preservation and utilization of structural information throughout the entire process.

## 3 Code Graph Construction

Before delving into the CGM, it is crucial to understand the repository-level code graph that CGM utilizes and the process of its construction. The primary aim of this code graph is to offer a structured representation of the structural and semantic information inherent in complex codebases.

We represent each repository as a directed graph $G = (V, E)$, where $V$ is the set of distinct entities in the codebase and $E$ is the set of edges between these entities. To be specific, the code graph includes up to seven types of nodes and five types of edges (details are provided in Appendix B). The node types vary in granularity, ranging from the repository level (REPO) to fine-grained attributes. The edge types comprise both hierarchical (i.e., contains) and reference dependencies (calls/imports/extends).

As shown in Figure 1, the hierarchical dependencies (i.e., the solid edges) span the code graph. In other words, all nodes are interconnected by edges reflecting hierarchical dependencies, establishing a top-down tree structure. This structure mirrors the organization of code entities as dictated by file systems and programming language syntax rules. Building this tree graph begins with AST parsing [52]. During this phase, code entities and their hierarchical dependencies are identified in a recursive manner: the root node (i.e., REPO) is added to the graph first, followed by its children (i.e., PKG and T-FILE), until all nodes without descendants (i.e., FUNC) are processed. With each recursion, directed edges are added from parents to children.

On the other hand, reference dependencies (i.e., the dashed edges) capture interactions between different entities, such as class inheritance, function calls, and module imports. Unlike hierarchical edges, which maintain a vertical hierarchy, reference edges create horizontal connections that may introduce cycles, such as those caused by recursive calls. These edges are typically not part of an AST. To derive them, we conduct a lightweight semantic analysis to resolve symbols, such as references or calls to classes and attributes. Once a target symbol is identified, an edge is added from the source node to the target node in the code graph.

Concerning node attributes, we retain the original content and line range of each node. This approach enables explicit graph traversal and retrieval and facilitates training models with enhanced semantic understanding capabilities. During post-processing, we remove the text contained in the child nodes from the parent nodes within the tree graph derived from the hierarchical dependencies. The resulting code graph is a text-rich graph [54] in which each node encapsulates a corresponding code snippet.

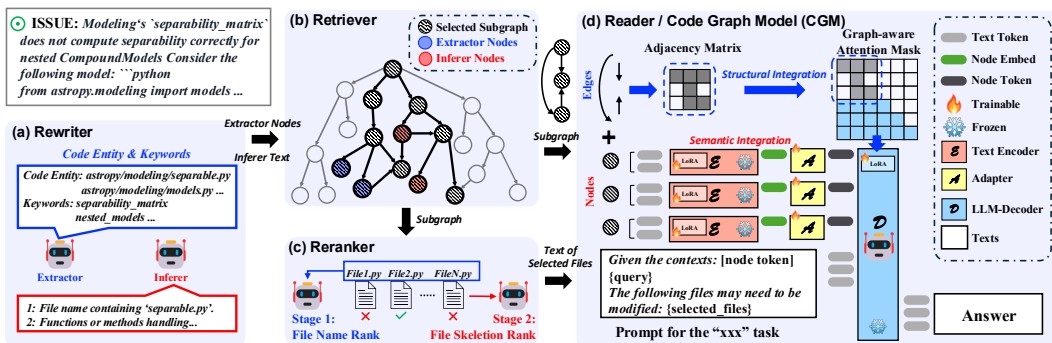

Figure 2: Architecture of CGM and its Graph RAG extension: Given an issue, (a) Rewriter extracts code entities and keywords from the issue (Extractor), and modifies the original issue into more detailed queries (Inferer). Based on this, (b) Retriever locates relevant nodes from the corresponding code graph; then expands these nodes to a connected subgraph by including neighboring and upstream nodes. (c) Reranker ranks retrieved results in two stages: File Name Rank and File Skeleton Rank, selecting the most relevant files for modification. Finally, (d) Reader (CGM) takes the retrieved graph and selected files as input. Each node's code content is encoded by an Encoder $\mathcal{E}$, producing a node token via the Adapter $\mathcal{A}$. Node tokens are then concatenated with text tokens in the prompt before entering the LLM decoder $\mathcal{D}$, where the adjacency matrix replaces its original attention mask.

# 4 Code Graph Models (CGMs)

In this section, we elaborate on the architecture of the Code Graph Model (CGM), the training strategy we adopted, and how we enhance the CGM via the Graph RAG framework.

## 4.1 Model Architecture

The architecture of the CGM is illustrated in Figure 2(d). CGM takes the code graph as inputs, enhancing the LLM's comprehension of both semantic and structural information within the graph. Below, we detail how CGM integrates both aspects into the LLM.

**Semantic Integration**: The code graphs are text-rich, with semantic information only residing in the textual contents of the nodes. As shown in Figure 2(d), we integrate the **node information** into the LLM decoder $\mathcal{D}$ by transforming node text into node tokens through an encoder $\mathcal{E}$ and an adapter $\mathcal{A}$.

Specifically for the encoder, we utilize the pretrained encoder from CodeT5+ [55], chosen for its proven effectiveness in processing both source code and text (comments and documentation). For nodes containing lengthy text, we segment the content into chunks of 512 tokens each. These chunks are processed independently by the encoder. To maintain graph consistency, we duplicate the source node for each chunk, preserving identical connections to other nodes. The chunks within a node are fully connected, and their sequential order is maintained through position embeddings in the LLM decoder $\mathcal{D}$. We fine-tune the encoder using Low-Rank Adaptation (LoRA) [17] to optimize its performance for downstream tasks.

The adapter $\mathcal{A}$ serves as a bridge between the encoder and LLM, projecting encoder outputs into the LLM's input embedding space. Following successful practices in Vision Language Models (VLMs) [56, 57], we implement the adapter as a two-layer MLP with GELU activation [58]. The adapter is trained from scratch with random initialization.

Unlike VLMs, which bridge different modalities, CGM's encoder $\mathcal{E}$ and decoder $\mathcal{D}$ are of the same modality, simplifying the alignment process. Furthermore, we compress each 512-token chunk (shown as gray tokens in Figure 2(d)) into a single node token (black tokens in Figure 2(d)) for the LLM decoder. This compression effectively extends the LLM's context length by a factor of 512, enabling the processing of extensive code repository contexts. Similar techniques, referred to as soft prompt compression, have been shown to enhance long-context modeling in recent studies [59, 60, 61].

**Structural Integration**: Besides node information, another challenge is integrating the **graph structure** into the LLM decoder $\mathcal{D}$. While LLMs excel at processing sequential data, they are not inherently designed to capture graph structures [54]. Traditional approaches have attempted to incorporate repository-level structural information by simply linearizing code snippets into sequences [14, 49], but this transformation often fails to preserve the explicit relationships between code entities.

To better maintain structural relationships, we introduce a **graph-aware attention mask** to replace the causal attention mask solely between node tokens in the LLM. This mask is derived from the code graph's adjacency matrix, taking into account the node duplication process described earlier. We then fine-tune the LLM with LoRA to adapt it to both the new attention mechanism and the node tokens from the adapter $\mathcal{A}$. This approach ensures that attention occurs only between neighboring nodes in the code graph, mimicking the message passing mechanism frequently used in spatial GNNs [62, 63].

## 4.2 Training Strategies

Given the pretrained encoder $\mathcal{E}$ and decoder $\mathcal{D}$, the training of the CGM consists of two main phases:

**Subgraph Reconstruction Pre-training:** This phase focuses on training the CGM to effectively capture both the semantic and structural aspects of code graphs. To achieve this, we introduce a novel pre-training task that requires the model to reconstruct code content from its corresponding code graph, a process we refer to as Graph-to-Code.

In this task, the inputs are subgraphs randomly sampled from large-scale code graphs, with a limited number of nodes. This constraint ensures that the corresponding output code remains below 8,000 tokens, allowing for computational efficiency and manageable context sizes during training. To enhance the meaningfulness of the output code, we employ a hierarchical approach that preserves the inherent dependencies within the code graphs as they are translated into text. Concretely, for higher-level nodes (e.g., REPO and PACKAGE), we position them at the beginning of the output sequence or their respective files to maintain hierarchical consistency. We then utilize the approach from DeepSeek-Coder [14] to perform topological sorting on all file nodes, thereby establishing a structured order for the code content. Lastly, intra-file nodes (e.g., CLASS and FUNCTION) are sorted by line numbers and concatenated within their respective files, culminating in a coherent text sequence that accurately represents the sampled subgraph.

**Noisy Fine-tuning:** This phase fine-tunes CGM on real-world issue-patch pairs [5], adapting it to practical software debugging and code editing tasks. As displayed in Figure 2(d), the model learns to generate code patches based on two inputs: (i) a subgraph and (ii) a text prompt that indicates the "oracle" files—files that require modification according to the ground-truth patch. The subgraph is constructed by combining the oracle files, their downstream nodes, and one-hop neighbors from the repository-level code graph. To improve model robustness, we intentionally introduce noise into the prompts: 10% include an irrelevant file that doesn't require modification, while another 10% omit at least one oracle file. This controlled noise exposure helps the model better generalize to real-world scenarios where inputs may be incomplete or contain irrelevant information.

## 4.3 The Graph RAG Framework

This section presents our Graph RAG framework, a streamlined extension of CGM designed for automated resolution of real-world repository tasks. The framework consists of four core modules: Rewriter, Retriever, Reranker, and Reader (the proposed CGM). This compact architecture stands in contrast to the SOTA agentless method, which requires ten distinct steps [2].

As illustrated in Figure 2, the framework operates sequentially. First, Rewriter enhances the original issue description to help Retriever identify relevant nodes in the code graph. Retriever then constructs a connected subgraph using both lexical and semantic search techniques. This subgraph serves as input for both Reranker and Reader. Reranker analyzes the subgraph to identify the Top $K$ files likely to be modified. Finally, Reader (CGM) generates the code patch using both the subgraph from Retriever and the selected files from Reranker. Rewriter and Reranker are implemented by prompting the open-source Qwen2.5-72B-instruct [20], while the semantic search in Retriever utilizes the open-source CGE-Large model [64]. In Appendix D, we provide a case study on how CGM solve a specific issue from scratch. Meanwhile, we report the computational costs of our framework, including the cost of code graph construction, in Appendix C.4

**Rewriter** comprises two subcomponents: Extractor and Inferer, as illustrated in Figure 2(a). Extractor identifies key code elements from the user query, including file names, function names, and relevant keywords. Inferer then enriches the query's semantics by providing more detailed functional descriptions. The specific prompts for both components are detailed in Appendix G.

**Retriever** generates a connected subgraph from the code graph for subsequent modules. As shown in Figure 2(b), Extractor nodes (blue nodes) are first identified through string matching with the code elements and keywords extracted earlier. Next, Inferer nodes are located (red nodes) through semantic search, comparing the Inferer's output with each node's textual information. These anchor nodes are

Table 1: Performance comparison of **open source system** on SWE-bench Lite and Verified. CS-3.5 denotes Claude-3.5-Sonnet-20241022, DS-V3 represents DeepSeek-V3, Q2.5C-32B means Qwen2.5-Coder-32B and Q2.5-72B stands for Qwen2.5-72B-Instruct. The icons 🤠 and 🔒 denote **open and closed-source models**, respectively.

(a): SWE-bench Lite

| Method | LLM | Agent | % R | Rank | All |
| --- | --- | --- | --- | --- | --- |
| DARS Agent | 🔒CS-3.5 | Yes | 47.00 | 1 | 6 |
| **CGM-SWE-PY** | 🤠Q2.5-72B | No | **43.00** | 2 | 8 |
| Lingxi | NA | Yes | 42.67 | 3 | 10 |
| CodeAct-v2.1 | 🔒CS-3.5 | Yes | 41.67 | 4 | 11 |
| PatchKitty-0.9 | 🔒CS-3.5 | Yes | 41.33 | 5 | 12 |
| Composio SK | 🔒CS-3.5 | Yes | 41.00 | 6 | 14 |
| Agentless-v1.5 | 🔒CS-3.5 | No | 40.67 | 7 | 32 |
| Moatless | 🔒CS-3.5 | Yes | 39.00 | 8 | 19 |
| Patched.Codes | 🔒CS-3.5 | Yes | 37.00 | 9 | 20 |
| **CGM-Multi** | 🤠Q2.5-72B | No | **36.67** | 10 | 23 |
| AppMap | 🔒CS-3.5 | Yes | 36.00 | 11 | 24 |
| Agentless Lite | 🔒o3-mini | No | 32.33 | 13 | 31 |
| Agentless-v1.5 | 🔒GPT-4o | No | 32.00 | 14 | 32 |
| Moatless | 🤠DS-V3 | Yes | 30.67 | 16 | 35 |
| SWE-Fixer | 🤠Q2.5-72B | Yes | 24.67 | 26 | 51 |
| Lingma SWE-GPT | 🔒Q2.5-72B | No | 22.00 | 28 | 57 |

(b): SWE-bench Verified

| Method | LLM | Agent | % R | Rank | All |
| --- | --- | --- | --- | --- | --- |
| OpenHands | NA | Yes | 65.80 | 1 | 1 |
| PatchPilot-v1.1 | NA | NA | 64.60 | 2 | 5 |
| SWE-Agent | 🔒CS-3.7 | Yes | 62.40 | 3 | 10 |
| Agentless-v1.5 | 🔒CS-3.5 | No | 50.80 | 4 | 25 |
| **CGM-SWE-PY** | 🤠Q2.5-72B | No | **50.40** | 5 | 26 |
| Composio SK | NA | Yes | 48.60 | 6 | 31 |
| Agentless Lite | 🔒o3-mini | No | 42.40 | 8 | 39 |
| Composio SK | 🔒CS-3.5 | Yes | 40.60 | 10 | 47 |
| SWE-Agent | 🤠Q2.5C-32B | No | 40.20 | 11 | 48 |
| Agentless-v1.5 | 🔒GPT-4o | No | 38.80 | 12 | 50 |
| SWE-Fixer | 🤠Q2.5-72B | Yes | 32.80 | 14 | 54 |
| Lingma SWE-GPT | 🤠Q2.5-72B | No | 30.20 | 15 | 58 |
| Lingma | 🤠Q2.5-72B | Yes | 28.80 | 16 | 60 |
| Lingma SWE-GPT | 🤠Q2.5-72B | No | 25.40 | 18 | 64 |
| SWE-Agent | 🔒GPT-4o | Yes | 23.20 | 16 | 67 |
| SWE-Agent | 🔒GPT-4 | Yes | 22.40 | 17 | 68 |

then expanded to include their one-hop neighbors, capturing local programming dependencies [65]. To ensure connectivity and incorporate upstream information, these expanded nodes are connected to the Root node (REPO in Figure 1). Finally, each FILE node in the subgraph is expanded to include all its internal nodes, aligning with Reranker's file-level output. The result is a repository-enhanced context subgraph representing the user query, asdenoted by the shaded nodes in Figure 2(b).

**Reranker** further refines the subgraph generated by Retriever, selecting only the Top $K$ files deemed most likely to be revised. This refinement is necessary because Retriever's output includes files that may only be referenced and not modified. Reranker operates in two steps: first, it selects $K = 10$ files based on the original user query and file names; next, it narrows this selection down to $K = 5$ files by individually scoring each one according to how relevant its file skeleton [2] is to the user query. The specific prompt for Reranker can be found in the Appendix G.

**Reader** receives two inputs: the subgraph from Retriever as node tokens (black tokens) and the selected files with their full contents as text tokens (gray tokens), as depicted in Figure 2(d). These inputs are combined using the prompt template in the white box on the left of the figure. The graph and text tokens complement each other by providing global and local information related to the user query. Using this comprehensive information, Reader (i.e., the CGM) generates the final response.

## 5 Experiments

In this section, we assess the performance of the CGM on two primary tasks: repository-level issue resolution and code completion, for both Python and Java programming languages. We also conduct a series of ablation studies to validate the effectiveness of the model design and training strategies.

### 5.1 Repository-Level Issue Fixing

This section evaluates the proposed CGM against other SOTA methods in resolving real-world software issues. We use three benchmark datasets: SWE-bench Lite, containing 300 issues from 11 Python repositories, SWE-bench Verified, containing 500 issues from 12 Python repositories, and SWE-bench-java Verified, comprising 91 issues from 6 Java repositories. All benchmarks utilize developer-written unit tests to verify the correctness of model-generated patches, ensuring rigorous evaluation. Performance is measured using the resolution rate (% R), defined as the percentage of successfully resolved issue instances. We present results for two variants of our model: **CGM-Multi**, trained for both issue resolution and code completion tasks across Python and Java repositories, and **CGM-SWE-PY**, specifically optimized for Python issue resolution. Detailed information regarding the datasets and implementations can be found in Appendix C.5.

As shown in Table 1(a), our CGM-SWE-PY model achieves a 43% resolution rate on SWE-bench Lite, placing it **first** among methods utilizing open-source models, **second** among those that implement open-source methods but use closed-source models, and **eighth** overall. Notably: (i) When compared to other methods based on open-source models, **CGM-SWE-PY outperforms Moatless+DeepSeek-**

Table 2: Performance evaluation on SWE-bench-java Verified. DS-V2 denotes DeepSeek-Chat-V2, DSC-V2 represents DeepSeek-Coder-v2, GPT-4o refers to GPT-4o-2024-05-13, DB-128K stands for Doubao-Pro-128k, and GPT-4o-MINI indicates GPT-4o-MINI-2024-07-18. The icons 👑 and 🔒 denote open-source and closed-source methods or models, respectively.

| Method | LLM | Agent | % R | Rank |
|---|---|---|---|---|
| 👑CGM-Multi | 👑Q2.5-72B | No | **14.29** | 1 |
| 👑SWE-agent | 👑DS-V2 | Yes | 9.89 | 2 |
| 👑SWE-agent | 👑DSC-V2 | Yes | 7.69 | 3 |
| 👑SWE-agent | 🔒GPT-4o | Yes | 6.59 | 4 |
| 👑SWE-agent | 🔒DB-128K | Yes | 1.10 | 5 |
| 👑SWE-agent | 🔒GPT-4o-MINI | Yes | 1.10 | 6 |

Table 3: Performance comparison on CrossCodeEval and ComplexCodeEval benchmarks. DeepSeek-236B represents DeepSeek-V2.5-236B, Mixtral-123B denotes Mistral-Large-Instruct-2411, and Qwen2.5-72B refers to Qwen2.5-72B-Instruct. Baseline models are evaluated using FIM (Fill-in-Middle) and one-hop expansion.

| Method | CrossCodeEval | | | | ComplexCodeEval | | | |
| | Java | | Python | | Java | | Python | |
| | EM | ES | EM | ES | EM | ES | EM | ES |
|---|---|---|---|---|---|---|---|---|
| Mixtral-123B | 47.17 | 82.23 | 53.92 | 82.42 | 37.00 | 64.81 | 31.00 | 62.48 |
| DeepSeek-236B | 44.74 | **83.81** | 58.54 | **85.03** | 36.00 | 63.08 | 32.00 | 60.60 |
| Qwen2.5-72B | 37.31 | 78.78 | 58.50 | 81.56 | 26.00 | 54.14 | 28.00 | 57.12 |
| CGM-Multi-72B | **50.21** | 80.76 | **61.20** | 84.30 | **47.00** | 78.86 | **43.00** | 72.60 |

**V3 by 12.33%** [15], despite DeepSeek-V3's generally superior performance in various coding benchmarks compared to our LLM decoder Qwen2.5-72B [21]. Furthermore, **it exceeds Lingma SWE-GPT by 21%**, even though the latter employs carefully curated COT (chain-of-thought) data to boost Qwen2.5-72B's effectiveness in issue resolution. (ii) In relation to other agentless frameworks, **CGM-SWE-PY slightly surpasses Agentless+Claude-3.5-Sonnet by 2.33% and significantly outperforms Agentless+GPT-4o by 11.00%**. This achievement is particularly noteworthy given that Agentless leverages a complex ten-step pipeline with more powerful closed-source models, while CGM-SWE-PY operates on a simpler four-step Graph RAG framework. **We attribute this success to CGM's enhanced capacity to interpret both semantic and structural information within repositories.** (iii) While the top methods on SWE-bench Lite are entirely closed-source regarding both models and implementations, CGM-SWE-PY's results are within 10% of these systems. This indicates that CGM-SWE-PY has the potential to compete with leading agent-based methodologies. **Compared to other open-sourced model-based methods, CGM significantly narrows the gap between open-source models and closed-source methods in issue-fixing scenarios.** (iv) Our multi-task model, CGM-Multi, achieves a resolution rate of 36.67% on SWE-bench Lite, ranking 23rd overall. The relatively lower performance compared to CGM-SWE-PY can be attributed to its broader focus, which encompasses both issue fixing and code completion tasks across Python and Java repositories. (v) We further apply CGM-SWE-PY to a larger Python benchmark—SWE-bench Verified in Table 1(b), where CGM-SWE-PY ranks **first** again among open weight models, and **fifth** among methods with open-source system.

In the SWE-bench-java evaluation for Java repositories as shown in Table 2, CGM-Multi records a resolution rate of 14.29%, significantly outclassing SWE-Agent build upon both closed-source and open-source models. These findings further substantiate the effectiveness of our proposed GCM and the specially designed Graph RAG framework.

## 5.2 Repository-Level Code Completion

In this section, we evaluate the CGM's performance on code completion tasks at the repository level for both Python and Java programming languages. Our evaluation uses two benchmarks: CrossCodeEval and ComplexCodeEval. Concretely, CrossCodeEval focuses on cross-file code completion, while ComplexCodeEval encompasses more intricate tasks, including API recommendations and test case generation. Performance is measured using two metrics: Edit Match (EM) and Edit Similarity (ES), evaluating how similar the generated code is to the ground-truth code. Detailed information regarding the datasets, metrics, and the implementation of baseline models can be found in Appendix C.6.

Table 4: Comparison of CGM with RAG variants on CrossCodeEval. Results are reported for Java and Python across multiple base models. Evaluation metrics include EM and ES.

| METHOD | CODELLAMA-7B | | | | DEEPSEEK-CODER-7B | | | |
| | JAVA | | PYTHON | | JAVA | | PYTHON | |
| | EM | ES | EM | ES | EM | ES | EM | ES |
|---|---|---|---|---|---|---|---|---|
| NORAG | 20.60 | 54.50 | 13.70 | 44.10 | 24.20 | 59.30 | 19.40 | 52.50 |
| BM25 | 23.42 | 66.13 | 21.76 | 69.09 | 22.49 | 66.78 | 23.30 | 70.84 |
| REPOFUSE | / | / | 24.80 | 71.05 | / | / | 27.92 | 73.09 |
| RLCODER | 26.23 | 67.61 | 26.60 | 72.27 | 26.09 | 67.31 | 30.28 | **74.42** |
| R2C2 | 35.60 | 58.50 | 23.60 | 42.90 | 41.60 | 64.60 | 32.70 | 54.00 |
| **CGM-MULTI** | **36.42** | **75.28** | **31.03** | **73.90** | **41.65** | **74.76** | **33.88** | 71.19 |

| METHOD | STARCODER-7B | | | | QWEN2.5-CODER-7B | | | |
| | JAVA | | PYTHON | | JAVA | | PYTHON | |
| | EM | ES | EM | ES | EM | ES | EM | ES |
|---|---|---|---|---|---|---|---|---|
| NORAG | 21.60 | 55.90 | 17.00 | 49.50 | 37.31 | 78.78 | 33.63 | 73.19 |
| BM25 | 22.16 | 67.80 | 22.33 | 69.60 | 49.37 | 82.63 | 43.15 | 78.66 |
| REPOFUSE | / | / | 24.20 | 70.82 | / | / | / | / |
| RLCODER | 24.73 | 69.08 | 25.82 | **72.11** | / | / | / | / |
| R2C2 | **38.10** | 63.60 | 30.90 | 51.90 | / | / | / | / |
| **CGM-MULTI** | 37.44 | **73.77** | **31.00** | 71.66 | **51.61** | **84.62** | **46.23** | **82.16** |

Table 5: Impact of each RAG (Retrieval-Augmented Generation) component on the performance of CGM for issue fixing. Results are reported as the resolve rate (% R) on SWE-bench Lite, demonstrating the contribution of rewriter, retriever, and reranker modules.

| SETTING | % R |
|---|---|
| - W/O REWRITER | 34.67 |
| - W/O RETRIEVER | 31.67 |
| - W/O RERANKER | 18.33 |
| - W/O R3 | 9.67 |
| - W/O CGM READER (FLATGRAPH) | 5.33 |

Table 3 presents the results for CGM-Multi, which uses Qwen2.5-72B-instruct as its LLM decoder. We compare it with similarly sized large language models, including Mistral-Large-Instruct-123B, DeepSeek-V2.5-236B, and the standalone Qwen2.5-72B-instruct. For all models, context retrieval for code completion is performed by identifying one-hop neighbors of the target file (that requires completion) in the code graph. While CGM-Multi processes the entire subgraph as input, baseline models only receive the textual content from the nodes. Results show that CGM-Multi performs on par with or exceeds other models on CrossCodeEval. **More importantly, it greatly outperforms the baseline models on ComplexCodeEval, demonstrating its superior capability in handling complex tasks through comprehensive subgraph analysis.**

Next, we evaluate CGM against other RAG methods for CrossCodeEval. The comparison includes several established systems: BM25, the default retrieval method in CrossCodeEval [66]; RLcoder [67], which employs reinforcement learning for retrieval optimization; RepoFuse [68], which integrates code graphs during retrieval but converts retrieved code snippets into linear text sequences; and R2C2 [69], which combines retrieved code snippets with Tree-sitter-generated abstract context as the input to the LLM. In our CGM implementation, we still construct input subgraphs by combining target files with their one-hop neighbors. We evaluate these methods using various base models for generation, including CodeLlama-7B, StarCoder-7B, DeepSeek-Coder-7B, and Qwen2.5-Coder-7B. This diverse set of comparison methods enables a comprehensive evaluation of CGM's effectiveness in long-context retrieval and understanding. As shown in Table 4, **CGM typically outperforms other RAG methods, regardless of the base model used, suggesting that graph-based context retrieval is more effective for code completion tasks.** Moreover, CGM's superiority over RepoFuse, which also uses code graphs for retrieval, can be attributed to CGM's explicit integration of structural information within the subgraph, whereas RepoFuse flattens node context into text sequences, obscuring the explicit dependencies among code entities.

### 5.3 Ablation Studies
In this section, we present key findings from our ablation studies, with detailed analysis available in Appendix C.7. Our investigation reveals four crucial insights: (i) **Graph RAG**: Our assessment of

Table 6: Impact of training strategies on CGM's performance. Results are reported for CrossCodeEval (Java and Python) in terms of EM and ES. Here, $\mathcal{E}$ denotes the encoder, $\mathcal{A}$ denotes the adapter, $\mathcal{D}$ denotes the LLM, and "combined" refers to the full CGM setup (w/ mask, w/ Recon, $\mathcal{A}$ + LoRA w/ $\mathcal{E}$ + $\mathcal{D}$).

| SETTING | CROSSCODEEVAL | | | |
| | JAVA | | PYTHON | |
| | EM | ES | EM | ES |
|---|---|---|---|---|
| MODULE | | | | |
| - FREEZE | 17.91 | 58.28 | 11.78 | 51.60 |
| - $\mathcal{A}$ | 41.70 | 77.25 | 34.90 | 74.54 |
| - LORA W/ $\mathcal{D}$ | 46.84 | 82.06 | 38.76 | 76.02 |
| - $\mathcal{A}$ + LORA W/ $\mathcal{D}$ | 49.51 | 83.21 | 43.15 | 79.84 |
| - W/O MASK | 48.71 | 83.40 | 42.21 | 80.46 |
| - W/O RECON | 42.78 | 80.29 | 39.77 | 75.87 |
| - COMBINED | **51.61** | **84.62** | **46.23** | **82.16** |

Table 7: Performance of CGM with different LLM backbones. Results are reported as the resolve rate (% R) on SWE-Bench Lite, demonstrating the model's ability to generalize across various sizes and architectures.

| BACKBONE | SWE-BENCH LITE (% R) |
|---|---|
| CGM | |
| - QWEN2.5-72B-INSTRUCT | 43.00 |
| - LLAMA3.1-70B-INSTRUCT | 25.33 |
| - QWEN2.5-CODER-32B-INSTRUCT | 28.67 |
| - QWEN2.5-CODER-7B-INSTRUCT | 4.00 |

Table 8: Analysis of the test-time scaling (TTS) performance using the Pass@$K$ metric. Results are reported as the resolve rate (% R) on both SWE-Bench Lite and SWE-Bench Verified, demonstrating the benefits of leveraging additional test-time computation.

| PASS@$K$ | SWE-BENCH LITE (% R) | SWE-BENCH VERIFIED (% R) |
|---|---|---|
| $K$=1 | 43.00 | 50.40 |
| $K$=2 | 44.33 | 51.40 |
| $K$=3 | 46.67 | 53.20 |

the Graph RAG modules in Table 5 shows that the presence of Rewriter, Retriever, and Reranker is essential for achieving optimal performance on the SWE-bench Lite benchmark. Notably, Reranker plays a pivotal role as it dictates which files should be modified. (ii) **Semantic Integration**: Joint fine-tuning of all three components in Table 6—encoder $\mathcal{E}$, the adapter $\mathcal{A}$, and the decoder $\mathcal{D}$—yields superior performance compared to keeping any component fixed. (iii) **Structural Integration**: The integration of graph structural information through attention masking is essential for optimal performance. (iv) **Training Strategies**: The subgraph reconstruction task, as described in Section 4.2, significantly contributes to improving the CGM's overall performance. (v) **Backbone Generalization**: Moreover in Table 7, CGM can also be generalized on backbones with different sizes, demonstrating its potential for resource-constrained scenarios. (vi) **Test-Time Scaling**: As detailed in Table 8, the test-time scaling strategy implemented via Pass@$K$ sampling significantly improves the performance of CGM on both SWE-Bench benchmarks.

## 6   Conclusion

In this paper, we present CGM, a novel graph-enhanced LLM architecture designed for comprehensive repository-level code understanding. By seamlessly integrating both semantic and structural information from codebases through a specialized encoder-adapter framework and graph-aware attention mechanisms, CGM demonstrates that sophisticated agent-based approaches and closed-source models are not necessarily required for complex SE tasks. When combined with our custom-designed Graph RAG framework, CGM achieves a remarkable 43.00% resolution rate in real-world issue-fixing scenarios on SWE-bench Lite, using only open-source models. Our work establishes a new direction for developing powerful, transparent, and accessible tools for automated SE.

# 7 Acknowledgement

This work was supported by Ant Group Research Intern Program. Prof. Haipeng Zhang was supported by Science and Technology Commission of Shanghai Municipality (25ZR1401256).

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

# A    Case in Issue Fix Scenario

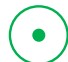

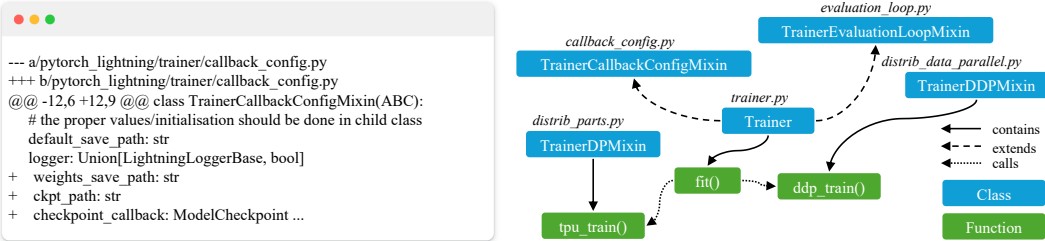

Figure 3: Illustration of a real-world issue from pytorch-lightning codebase, where a user wants to disable automatic checkpoint loading. Given the original issue, the corresponding diff-formatted patch (bottom left) shows all the code modifications in a linear fashion. Compared to code sequences, the relationships between them can be more clear if we represent the code as a graph (bottom right), where containment (solid lines), inheritance (dashed lines), and function calls (dotted lines) explicitly demonstrate the connections between different code snippets.

# B    Details of Code Graph

## B.1    Node and Edge Types in Code Graph

This section provides the node and edge types defined in our code graph. Table 9 and Table 10 detail the categories of nodes and edges, respectively. For now, code graph supports two objected-oriented programming language (Python and Java).

Table 9: Node Types in Code Graph.

| Node Type | Description |
|---|---|
| REPO | Virtual node, represents the root of the repository |
| PACKAGE | Virtual node, acts as a representation of a directory in the file system |
| FILE | Files ending with ".py" |
| TEXTFILE | Files that do not end with ".py", such as Markdown (.md) files and text (.txt) files |
| CLASS | In object-oriented programming, a class is a blueprint for creating objects |
| FUNCTION | Refers to the function within classes or standalone function |
| ATTRIBUTE | Includes global variables, member variables, and local variables |

Table 10: Edge Types in Code Graph.

| Edge Type | Description |
|---|---|
| contains | Indicating a hierarchical relationship in which one entity contains another entity |
| calls | This type of edge captures the dynamic invocation relationship |
| extends | Representing an inheritance relationship, where one class extends another class |
| imports | Represent dependencies where one file imports another class/function |
| implements | This edge is exclusively applicable to Java, denoting the relation where a class implements an interface |

## B.2    Handling of Complex Dependiences

During the construction of code graph, we explicitly address both dynamic calls and multiple inheritance in the following way.

**Dynamic Calls:** We employ a conservative resolution approach following the over-approximation principle [70]. When encountering base class method calls (e.g., Base.method()), we include all possible overriding implementations from subclasses in the calls set. This ensures we don't miss any potential execution paths.

**Multiple Inheritance:** We utilize the Class Hierarchy Analysis (CHA) algorithm [71] to properly handle inheritance relationships, including cases where classes inherit from multiple parent classes.

### B.3   Search on Code Graph

Graph search can be easily implemented in code graph. The first step usually begins with finding the source node. This can be achieved by many ways, such as indexing, keyword and embedding matching. Starting from the source node, different strategies can be applied, such as one-hop neighborhood, depth-first search (DFS), breadth-first search (BFS), random walk, etc. It is up to the application scenarios to decide which search algorithm is the best. The result of graph search could be a sub-graph of the whole repository-level graph, containing the most relevant context for specific problems.

## C   Implementation Details

### C.1   Details of Training CGM

This section details how we train CGM-Multi (multi-language version), CGM-SWE-PY (tailored for Python issue fixing), and CGM 7B series (based on different 7B base models).

#### C.1.1   Training Data

As mentioned in section 4, we construct training data for different training phase of CGM respectively. Meanwhile, to enhance the model's ability in code completion, we also construct code completion samples for fine-tuning of code completion task. To explore the performance of CGM across different programming languages, our data includes both Python and Java. When constructing the training data, we filter out the repositories involved in the test sets for testing to avoid data leakage.

**Data for Subgraph Reconstruction Pre-training**: We obtain 500k Python and 360k Java subgraphs (with the maximum length of 8k tokens) from a total of 20k high-star Github repositories.

**Data for Issue Fixing**: We collect 200k issue-patch pairs (100k per language), from GitHub pull-requests. Among the 100k Python pairs, 14k are sourced from the SWEBench training set [8].

**Data for Code Completion**: The code completion samples are self-constructed from the above repositories, 250k per language.

#### C.1.2   CGM-Multi

We initialize CGM-Multi with Qwen2.5-72B-Instruct [20] as the base LLM. Then pre-train it using subgraph reconstruction data and fine-tuning data (issue-fixing and code completion) in both two languages (Python and Java). To ensure balance between different languages, we use 360k subgraph reconstruction data for both languages. Training uses 4-bit QLoRA on 64 A100s (batch=32, lr=1e4, and epoch=2). The encoder combines CodeT5+ [55] with LoRA (rank=64, and alpha=32), and the adapter uses a two-layer MLP with GELU activation. The first layer of the adapter maps the CodeT5+ output dimension of 256 to 8192, and the second layer maintains the dimension of 8192, which aligns with the LLM's hidden dimension. We adopt Xformers [72] for efficient attention computation.

#### C.1.3   CGM-SWE-PY

CGM-SWE-PY, as a model specifically designed for SWE-bench Lite, is pre-trained using python subgraph reconstruction data (the entire 500k) and fine-tuned on specific python issue-fixing data (the 14k sourced from SWEBench training set). Besides, all details of training and parameters are set the same as CGM-Multi.

Table 11: Recall performance of each Graph RAG module on SWE-bench Lite and SWE-bench-java Verified. The table shows the recall percentage for Retriever, Reranker Stage 1, and Reranker Stage 2 components.

| MODULE | SWE-BENCH LITE % RECALL | SWE-BENCH-JAVA VERIFIED % RECALL |
|---|---|---|
| RETRIEVER | 94 | 87 |
| RERANKER STAGE 1 | 89 | 74 |
| RERANKER STAGE 2 | 87 | 60 |

### C.1.4 CGM 7B Series

We train several small-scale CGMs based on the existing 7B base models to compare with small-scale models on code completion benchmarks. Specifically, we trained CGMs in seperate language based on CodeLlama-7B, StarCoder-7B, DeepSeek-Coder-7B, and Qwen2.5-Coder-7B-Instruct, respectively. For each model in each language, we use training data in the target language during both pre-training and fine-tuning stages. For example, we train CodeLlama-7B with 500k Python subgraph reconstruction data and 250k Python code completion samples, and then evaluate it on the Python test set of crosscodeeval.

Except for modifying the parameters in LoRA (set rank=32, and alpha=16), other training/parameter settings are consistent with CGM-Multi.

### C.2 Recall Results for the Graph RAG Framework

We provide the recall of each component of our Graph RAG framework (in the file level), as shown in Table 11. The recall of each component on SWE-bench-java Verified are lower than those on SWE-bench Lite. One possible reason may be that the issues in SWE-bench Lite usually requires modifying one file, while the issues on the SWE-bench-java Verified sometimes need to modify multiple files.

### C.3 Hyperparameters for Inference

We use the same parameter settings for inference with LLMs (CGMs and Qwen2.5-72B-Instruct in the Graph RAG framework), setting $top_k = 20$, $top_p = 0.8$, temperature = 0.7, and repetition penalty = 1.1.

### C.4 Cost Analysis

In this section, we present a cost analysis of the overall process, including the time required for code graph construction and computational expenses.

### C.4.1 Code Graph Construction

The construction of a repository-level code graph usually takes 3 minutes or more depending on the complexity of the code repository (such as the implementations of different classes and the calling relationships between codes). Since code graph construction can be performed offline, it does not impact real-time inference workflows. Additionally, optimizations such as incremental updates and parallel processing can further reduce latency for large-scale repositories.

### C.4.2 Cost of Each Module

We analyze the runtime and resource requirements of each key module in our system, focusing on LLM inference overhead, memory consumption, and latency scaling.

**Rewriter**:

• Requires two sequential LLM calls (Qwen2.5-72B-Instruct).

**Retriever**:

- Anchor node matching and subgraph generation take 3–7 seconds per issue.
- Lightweight CPU operation.

**Reranker**:

- Requires two sequential LLM calls (Qwen2.5-72B-Instruct).
- Latency additive to Rewriter ($2\times$ single-call time).

**Reader (CGM)**:

- Table 12 reports the memory consumption and inference latency.
- Latency increases by 0.5–0.7s per 1k tokens (1k→8k: 3.9s→8.6s).

Table 12: Inference Time and Memory Cost of CGM (Qwen2.5-72B).

| # Input Tokens | Time (s) | Memory (GB) |
|---|---|---|
| 1,000 | 3.934 | 68.79 |
| 2,000 | 4.408 | 68.98 |
| 3,000 | 5.055 | 69.43 |
| 4,000 | 5.808 | 70.26 |
| 5,000 | 6.432 | 70.77 |
| 6,000 | 7.163 | 70.98 |
| 7,000 | 7.838 | 71.44 |
| 8,000 | 8.553 | 72.02 |

### C.5  Experimental Setup of Issue Fixing

#### C.5.1  Implementation Details of CGM

To adapt CGM in the issue-fix scenario, we extend CGM to a GraphRAG framework (as described in section 4). In this scenario, the inputs of CGM are the corresponding subgraph and prompt generated by R3 (Rewriter, Retriever, Reranker). In the experiment, we compare two pre-trained CGMs, CGM-Multi and CGM-SWE-PY (see Appendix C.1 for training details), as Reader in our Graph RAG framework.

#### C.5.2  Datasets

The following three benchmarks, which focus on repository-level issue fixing, are all evaluated using the Docker executable environment.

- **SWE-bench Lite** [5]: SWE-bench Lite contains 300 self-contained repository-level issues from 11 repositories, designed to test the model's understanding of repository-level code changes and its ability to generate correct patches primarily focused on Python. It provides a realistic software engineering environment, including execution contexts, to evaluate the model's ability to resolve real-world issues.

- **SWE-bench Verified** [73]: SWE-bench Verified contains 500 self-contained repository-level issues from 12 repositories. This dataset contains samples that have been verified to be non-problematic by human annotators.

- **SWE-bench-java Verified** [6]: This dataset include 91 Java issues from 6 repositories, enabling cross-language evaluation. Like SWE-bench Lite, it provides execution environments to validate the correctness of generated patches.

#### C.5.3  Evaluation Metrics

**Resolve Rate (% R):** The metrics used in the above benchmarks is Resolve Rate, which evaluates the correctness of generated patches for the issue-fix task. A patch is considered resolved if it correctly addresses the issue and is a superset of the ground-truth edits.

## C.6 Experimental Setup of Code Completion

### C.6.1 Implementation Details of CGM

As a simpler scenario than issue fixing, the files that need to be modified are given in the code completion tasks. Therefore, we obtain the input subgraph of CGM through a heuristic method, rather than the Graph RAG framework. To be specific, we take the given incomplete file as the center node, obtaining its one-hop ego graph from the repository-level code graph. Note that nodes that need to be completed are not considered in this process. The resulting subgraph (graph modalities), and the incomplete files (text modalities), form the inputs to the CGM.

In this experiment, we use two size of CGMs for evaluation. Training details for the large-scale CGM-Multi (72B) and small-scale CGM 7B series can be found in the Appendix C.1.

### C.6.2 Datasets

- **CrossCodeEval** [66] is an emerging benchmark for cross-file code completion, which is constructed from a wide range of real-world repositories from GitHub in four popular programming languages: Python, Java, TypeScript, and C#. In our experiments, we evaluate the model's code completion ability on only two languages, Java and Python. As shown in Table 13, we provide the dataset statistics of the CrossCodeEval benchmark for Java and Python.

- **ComplexCodeEval** [74] is a new benchmark for evaluating the performance of large code models in complex development scenarios. It includes 3,897 Java samples from 1,055 Java code repositories and 7,184 Python samples from 2,107 Python code repositories. Following the original setup of this benchmark, we randomly selected 100 samples each in Python and Java for the evaluation. Table 14 and Table 15 show the information of the selected samples. When evaluating the code completion capability, unlike the original setup which requires the model to complete the second half of the function, we ask the model to complete the middle line of the function given the contextual information.

Table 13: The statistics of the CrossCodeEval for Java and Python.

| Feature | Java | Python |
|---|---|---|
| # Repositories | 239 | 471 |
| # Files | 745 | 1368 |
| # Examples | 2139 | 2665 |

### C.6.3 Evaluation Matrics

When evaluating a prediction code $y$ in comparison to the reference ground truth $y^*$, the above benchmarks utilize the following two metrics: the exact match accuracy (EM) and the Levenshtein edit similarity (ES).

- **EM**: The exact match accuracy (EM) is determined by an indicator function. This function takes a value of 1 when the prediction $y$ is exactly equal to the reference $y^*$, and 0 otherwise.

- **ES**: The Levenshtein edit similarity (ES) is calculated using the formula

$$ES = 1 - \frac{\text{Lev}(y, y^*)}{\max(\|y\|, \|y^*\|)}. \tag{1}$$

Here, $\|\cdot\|$ is used to compute the length of a string, and Lev() is employed to calculate the Levenshtein distance between the two strings $y$ and $y^*$.

### C.6.4 Baselines: Base Model

Given the subgraph same to CGM (see Appendix C.6.1), we textualize the graph into text sequnce and input into the following baselines based on the prompt templates provided by each benchmark [66, 74]. Since these base models have limitations on context length, we perform truncation on text inputs larger than 8k.

**Mistral-Large-2** [75] is a model developed by Mistral AI with 123 billion parameters. It stands out with a remarkable 128k tokens context length, proficiently handling dozens of languages and over 80 programming languages, excelling in code generation, mathematics, and reasoning. To be specific, we chose Mistral-Large-Instruct-2411 as the latest version of Mistral-Large to compare with CGM.

**DeepSeek-V2.5** [14] is a strong Mixture-of-Experts (MoE) language model characterized by economical training and efficient inference. It comprises 236B total parameters, of which 21B are activated for each token.

**Qwen2.5** [20] is a decoder-only LM series whose size varies from 0.5B to 72B trained on 18 trillion tokens. Its context length support up to 128K tokens and can generate up to 8K tokens.

**Qwen2.5-Coder** [29] is a series of code-specific language models developed by Alibaba. Derived from Qwen2.5, it comes in six sizes, is trained on a vast 5.5-trillion-token corpus, and excels in various code-related tasks, outperforming many models of the same or larger size. Since it uses a specific format and tokens for training on the fill-in-middle code completion task, we followed this prompt setting during the evaluation to obtain its true performance.

For the inference of the baseline model, we all deployed the above models using the VLLM framework with the model's default settings. All models inference on 4 A100s with 80G VRAM, except for DeepSeek-V2.5, which requires 8 A100s.

### C.6.5 Baselines: RAG Method

**BM25** [76] is a classic information-retrieval algorithm based on the probabilistic model. Its core idea is to rank documents based on the relevance between query and documents. It serves as a traditional retrieval method that does not regard the structural information naturally existing in the coding task, and only performs similarity matching based on word frequency and text length. It was used in the original CrossCodeEval dataset to search for cross-file information based on the code snippets. In our experiments, we directly use the BM25 results provided by CrossCodeEval.

**R2C2-Coder** [69] is a method that aims to enhance and benchmark the real-world repository-level code completion abilities of code Large Language Models. In particular, $R^2C^2$-Enhance reduces cross-file information to skeleton[5] text by syntactically analyzing the content of code files. The cross-file context is retrieved using BM25 after forming a candidate retrieval pool together with the context obtained from semantic-based retrieval. It takes into account structural information of the code but does not establish graph relations across code files.

**RepoFuse** [68] is a solution for the Context-Latency Conundrum in repository-level code completion. It constructs Code Knowledge Graph by analyzing the code graph dependencies in the repository and uses the repository-level graphs for retrieval. It integrates the Rationale Context obtained by analyzing the repository code structure and the Analogy Context based on the retrieval of similar code blocks, and filtering the context by scoring function.

**RLCoder** [67] is a reinforcement-learning-based framework for repository-level code completion, which can effectively improve code completion performance and has good generalization ability. During training, the RLRetriever is trained with a reward mechanism based on weighted perplexity to learn retrieval, while a stop-signal mechanism is introduced to filter candidate codes. In inference, the trained RLRetriever retrieves useful candidate codes from the code repository and inputs them together with the incomplete code into the generator to complete code generation.

### C.7 Details of Ablation Study

In this section, we first conduct an ablation study on our Graph RAG framework to verify the effectiveness of each component by SWE-bench Lite (Table 5). Then, we conduct the other ablation study on CGM itself to evaluate the effectiveness of model design by CrossCodeEval dataset (Table 6).

---

[5]The file skeleton is a hierarchical structure of the contents of a code file, containing class and function declarations, without specific definitions and comments.

### C.7.1 Variants of Graph RAG Framework

The Graph RAG framework, comprising Rewriter, Retriever, Reranker, and Reader, extends CGM to real-world issue fixing. In Table 5, we verify the effectiveness of each component in our Graph RAG framework by removing them. Here, we use CGM-SWE-PY (see Appendix C.1 for training details) as Reader.

- **w/o Rewriter:** We directly perform semantic search based on the original issue descriptions, obtain the anchor nodes from the code graph, and provide them to Retriever. Removing Rewriter results in an 8.33% performance drop, which proves its effectiveness in enhancing the original issue descriptions.

- **w/o Retriever:** Since there is no Retriever to provide filtered files and subgraphs, we input all the files in the original codebase into Reranker's Stage 1 for selection, and at the same time append the key information output by Rewriter into Reranker's prompts. Based on the files output by Reranker, we build a subgraph using these files and their one-hop neighbors, as the graph modality input of CGM. The exclusion of Retriever results in an 11.33% performance degradation, a more severe drop than removing Rewriter, highlighting its importance in providing issue-related subgraph.

- **w/o Reranker:** We use the top 5 files that are most similar to the query in embedding space (during semantic search) from the FILE node obtained by Retriever and provide them to Reader as the files to be modified. Removing Reranker results in the largest performance drop (decreased by -24.67%), emphasizing its importance in improving the precision of retrieval results and providing the right, relevant files to Reader.

- **w/o R3:** To evaluate the effectiveness of the RAG module, we create a baseline which removes the first three modules (Rewriter, Retriever, and Reranker) and feed the entire (truncated when the length exceeds the context length of the base model) repository graph as input to Reader during fine-tuning. Removing the RAG module leads to a poor performance (decreased by 33.33%), possibly due to excessive noise from the unfiltered repository graph and information loss from context-window truncation.

- **w/o CGM Reader (FlatGraph):** To verify the effectiveness of CGM Reader in jointly modeling semantics and structure, we create a naive graph-based baseline which flattens code snippets based on topological structure [28], representing an alternative Reader with structure-enhanced fine-tuning. The naive graph-based Reader only achieves 5.33% on SWE-bench Lite, far behind the proposed CGM (decreased by 37.67%).

### C.7.2 Variants of CGM

In Table 6, we compare CGM with its variants in the following three aspects. The CGM we use here is trained on Qwen2.5-Coder-7B Instruct (see Appendix C.1 for training details).

- **Semantic Integration:** To verify the design of CGM in understanding semantic information, we compare it with four types of variants: (1) freeze all parameters (include Encoder, Adapter, and LLM Decoder) (2) training the Adapter $\mathcal{A}$ (3) training the LLM Decoder $\mathcal{D}$ (4) training both the Adapter $\mathcal{A}$ and LLM Decoder $\mathcal{D}$. Table 6 demonstrates that training the adapter $\mathcal{A}$ alone leads to significant improvements in the EM performance: a 22.26% increase for Java and a 21.43% increase for Python when comparing CGM-$\mathcal{A}$ with GGM-Freeze. Additionally, further training the LLM decoder $\mathcal{D}$ in conjunction with the adapter $\mathcal{A}$ esults in further enhancements, yielding a 5.33% improvement for Java and a 4.80% improvement for Python. Finally, when the encoder $\mathcal{E}$, the adapter $\mathcal{A}$, and the decoder $\mathcal{D}$ are all trained together, we observe an additional increase of 5.71% for Java and 6.12% for Python. This data illustrates that fine-tuning the encoder $\mathcal{E}$, the adapter $\mathcal{A}$, and the decoder $\mathcal{D}$ is essential to effectively align the graph and code modalities.

- **Structural Integration:** To verify the design of CGM in integrating structural information, we remove the graph-aware attention mask during training, and use the original causal mask (denoted as "w/o MASK"). As shown in Table 6, substituting the graph-aware attention mask in the CGM with a standard causal mask results in a drop of 8.61% in EM performance for Java and 5.56% for Python. This demonstrates the necessity of incorporating the structural information from the code graph into the CGM to maintain optimal performance.

- **Training Strategy:** We remove the subgraph reconstruction pre-training task to verify the effectiveness of this task, denoted as "w/o RECON". Subgraph reconstruction pre-training plays a crucial role, contributing 7.65% to the overall EM improvements.

## C.8 Generalization of CGM on Different Backbones

To evaluate CGM with different backbones, we trained CGM using Llama3.1-70B-Instruct, Qwen2.5-Coder-32B-Instruct, and Qwen2.5-Coder-7B-Instruct, in addition to Qwen2.5-72B. The results are summarized in Table 7.

We find that the performance of CGM positively correlates with the LLM decoder's inherent coding and instruction-following abilities. For example, Llama3.1-70B-Instruct CGM's performance decreased 17.67% compared to Qwen2.5-72B, possibly due to weaker inherent coding abilities (see Table 2 in [20]). Still, it surpassed Lingma-SWEGPT [1] built on Llama3.1-70B-Instruct by 18.33%, demonstrating CGM's power in improving open-source LLMs.

## C.9 Test-Time Scaling Analysis

To further investigate the impact of inference-time computation, we analyze the performance of CGM under the test-time scaling (TTS) strategy, using the standard Pass@$K$ metric. This approach generates $K$ independent solutions for each issue and considers the issue resolved if at least one of the solutions passes the unit tests.

The results are summarized in Table 8. We observe that increasing the number of attempts ($K$) leads to a consistent and substantial improvement in the resolve rate (% R) across both benchmarks. Specifically, by increasing $K$ from 1 (no scaling) to 3, the performance on SWE-Bench Lite rises from 43.00% to 46.67%, an improvement of 3.67%. Similarly, on the more challenging SWE-Bench Verified subset, the resolve rate increases from 50.40% to 53.20%, a gain of 2.80%. This analysis demonstrates that allocating additional compute during inference through parallel sampling significantly enhances the model's ability to generate and select a correct solution for complex software engineering tasks, confirming the benefits of leveraging this robust decoding strategy.

Table 14: The Repositories and funcitons selected from ComplexCodeEval-Python.

| Repository | Function |
| --- | --- |
| IntelLabs/coach | validate_output_action_space |
| scikit-learn-contrib/category_encoders | transform |
| boto/boto3 | document_collections |
| flink-extended/ai-flow | get_conn |
| indico/indico | _process |
| aleju/imgaug | _generate_intersection_points |
| lucyparsons/OpenOversight | send_email |
| williamfzc/stagesepx | load_frames |
| dj-stripe/dj-stripe | _resync_instances |
| biosustain/potion | parse_request |
| MLBazaar/BTB | _fit |
| mljar/mljar-supervised | from_json |
| archesproject/arches | save |
| uber/causalml | causalsens |
| digiteinfotech/kairon | request |
| DeepLabCut/DeepLabCut | interpolate |
| WeblateOrg/weblate | check_component |
| oxan/djangorestframework-dataclasses | to_internal_value |
| etsy/boundary-layer | load |
| grafana/oncall | authenticate |
| trypromptly/LLMStack | process |
| weihuayi/fealpy | grad |
| django-cas-ng/django-cas-ng | get |
| lociii/jukebox | index |
| LAMDA-NJU/Deep-Forest | fit_transform |
| jazzband/django-simple-history | history_form_view |
| fabfuel/ecs-deploy | assume_role |
| waterdipai/datachecks | log |
| pfnet/pfrl | select_action |
| bhch/django-jsonform | render |
| allenai/OLMo | sample_nodes |
| AI4Finance-Foundation/ElegantRL | init_before_training |
| someengineering/fixinventory | parse_args |
| ssube/onnx-web | run |
| IntelAI/nauta | create_tensorboard |
| scikit-learn/scikit-learn | fit |
| awslabs/aws-embedded-metrics-python | probe |
| amundsen-io/amundsen | init |
| DataCanvasIO/DeepTables | fit |
| diyan/pywinrm | build_session |
| adamchainz/django-perf-rec | set_and_save |
| ihmeuw-msca/CurveFit | fit |
| google-research/weatherbench2 | compute |
| langroid/langroid | load |
| jina-ai/jcloud | _get_post_params |
| tfeldmann/organize | from_string |
| georgia-tech-db/evadb | exec |
| sibson/redbeat | is_due |
| bread-and-pepper/django-userena | process_request |
| betodealmeida/shillelagh | supports |
| kakaoenterprise/JORLDY | sample |
| openstack/neutron | get_total_reservations_map |
| mobiusml/hqq | quantize |
| django-json-api/django-rest-framework-json-api | get_paginated_response |
| nasaharvest/presto | add_masked_tokens |
| locuslab/mpc.pytorch | grad_input |

| | |
|---|---|
| Lightning-Universe/lightning-flash | transform |
| openxrlab/xrlocalization | knn_ratio_match |
| bentoml/BentoML | from_yaml_file |
| bayesiains/nflows | inverse |
| open-mmlab/mmcv | _resize |
| threat9/routersploit | run |
| hscspring/hcgf | train |
| martenlienen/torchode | from_k |
| arthurmensch/modl | split |
| pyg-team/pytorch-frame | forward |
| DjangoGirls/djangogirls | save |
| DataCanvasIO/Hypernets | create |
| randovania/randovania | format |
| materialsproject/fireworks | run_task |
| LinkedInAttic/naarad | generate |
| gift-surg/NiftyMIC | read_similarities |
| Project-MONAI/MONAILabel | entropy_3d_volume |
| griffithlab/pVACtools | execute |
| Giskard-AI/giskard | run |
| Zero6992/chatGPT-discord-bot | get_cookie_list |
| intelligent-machine-learning/dlrover | _save |
| florimondmanca/djangorestframework-api-key | save_model |
| GhostManager/Ghostwriter | clean |
| allwefantasy/auto-coder | merge_code |
| caktus/django-treenav | save |
| simpeg/simpeg | eval_deriv |
| arcee-ai/mergekit | _make_schedule |
| alex-petrenko/sample-factory | _save |
| RoboSats/robosats | submit_payout_address |
| pallets/quart | _create_request_from_scope |
| michael-lazar/rtv | get_mimetype |
| aurelio-labs/semantic-router | from_file |
| drivendataorg/deon | read |
| element-hq/synapse | generate_config_section |
| aquasecurity/kube-hunter | is_aws_pod_v2 |
| CarterBain/AlephNull | simulate |
| metauto-ai/GPTSwarm | optimize_swarm |
| ml6team/fondant | write_dataframe |
| pytorchbearer/torchbearer | save_checkpoint |
| intelowlproject/IntelOwl | _subquery_weight_org |
| chainer/chainerrl | initialize |
| petuum/adaptdl | optimize |
| regel/loudml | forecast |
| ansible/ansible | construct_mapping |

Table 15: The Repositories and funcitons selected from ComplexCodeEval-Java.

| Repo | Function |
|---|---|
| apache/tajo | findScalarFunctions |
| spring-projects/spring-batch | afterPropertiesSet |
| tencentmusic/supersonic | addAliasToSql |
| tmobile/pacbot | listAssets |
| microcks/microcks | createGenericResourceService |
| jtalks-org/jcommune | showNewQuestionPage |
| spring-projects/spring-data-redis | executeWithStickyConnection |
| apache/james-project | from |
| apache/hop | getXml |

| | |
|---|---|
| apache/incubator-dolphinscheduler | expandListParameter |
| apache/archiva | commit |
| Alfresco/alfresco-repository | check |
| 52North/SOS | init |
| kubernetes-client/java | index |
| xwiki/xwiki-platform | getFileItems |
| ctripcorp/x-pipe | analyze |
| digital-preservation/droid | getAvailableSignatureFiles |
| IridiumIdentity/iridium | generate |
| sofastack/sofa-acts | parseGenTableDatas |
| ProgrammeVitam/vitam | switchIndex |
| revelc/formatter-maven-plugin | init |
| Hack23/cia | unmarshallXml |
| immutables/immutables | oneLiner |
| pentaho/pentaho-platform | startup |
| ORCID/ORCID-Source | getWorkInfo |
| 88250/latke | resolve |
| mybatis/guice | get |
| GoogleCloudDataproc/spark-bigquery-connector | hashCode |
| gbif/ipt | add |
| jhy/jsoup | submit |
| neo4j/neo4j | nodeApplyChanges |
| PaladinCloud/CE | getAssetLists |
| alibaba/SREWorks | execute |
| jenkinsci/plugin-installation-manager-tool | installedPlugins |
| apache/syncope | getAdminRealmsFilter |
| apache/hadoop | checkAllVolumes |
| Qihoo360/Quicksql | distinctList |
| openlookeng/hetu-core | updateRows |
| zanata/zanata-platform | getLocales |
| AutoMQ/automq | persistentVersionedKeyValueStore |
| OctoPerf/kraken | list |
| metamx/druid | run |
| kiegroup/optaweb-vehicle-routing | startSolver |
| oceanbase/odc | bind |
| lennartkoopmann/nzyme | recordFrame |
| Stratio/Decision | childEvent |
| alibaba/velocity-spring-boot-project | getMatchOutcome |
| Aiven-Open/klaw | getConsumerGroupDetails |
| apache/doris-manager | createTable |
| apache/shardingsphere-elasticjob | init |
| apache/rya | distinct |
| ixrjog/opscloud4 | queryMyWorkRole |
| google/nomulus | validateDomainName |
| koraktor/steam-condenser-java | rconExec |
| wikimedia/wikidata-query-rdf | load |
| techa03/goodsKill | getSeckillList |
| runelite/runelite | onChatMessage |
| jenkinsci/blueocean-plugin | validateAccessTokenScopes |
| MyCATApache/Mycat-Server | formatProperties |
| jenkinsci/gitea-plugin | getFileLink |
| gentics/mesh | getUid |
| twilio/twilio-java | fromHttpRequest |
| ppdaicorp/das | checkSql |
| insideapp-oss/sonar-flutter | define |
| dschulten/hydra-java | linkTo |
| alibaba/fastjson2 | of |
| opencast/opencast | multiTrimConcat |

| | |
|---|---|
| spring-projects/spring-data-jpa | removeSubqueries |
| jline/jline3 | open |
| star-whale/starwhale | list |
| javaparser/javaparser | solveSymbolInType |
| datavane/datasophon | syncUserToHosts |
| sakaiproject/sakai | upgradeRoleString |
| alswl/yugong | queryAndSaveToQueue |
| zanata/zanata-server | parseGlossaryFile |
| aliyun/aliyun-log-java-producer | tryAppend |
| google/mug | forDoubles |
| apache/druid | wrap |
| ExpediaGroup/styx | equals |
| apache/kylin | encrypt |
| dCache/dcache | map |
| Asqatasun/Asqatasun | findByAuditAndUrl |
| mybatis/mybatis-3 | register |
| apache/poi | setArrayFormula |
| mitreid-connect/OpenID-Connect-Java-Spring-Server | parse |
| dianping/puma | copyFromLocal |
| alturkovic/distributed-lock | refresh |
| twitter/hraven | getAppId |
| OpenOLAT/OpenOLAT | isSetOfFlashcardExisting |
| apache/rocketmq | addTransactionSubscription |
| RIPE-NCC/whois | parse |
| odpi/egeria | buildGlossaryTermContext |
| ShifuML/shifu | exec |
| ozimov/spring-boot-email-tools | mergeTemplateIntoString |
| NationalSecurityAgency/datawave | from |
| spring-projects/spring-data-cassandra | addProperty |
| opennetworkinglab/onos | parse |
| Graylog2/graylog2-server | authenticate |
| openmrs/openmrs-core | handle |
| webx/citrus | getFastConstructor |

## D    CGM for Issue Fixing: A Case Study

In this section, we take a real issue from the django/django repository as an example to show how CGM solves a specific problem. Figure 4 provides the original issue description and the intermediate outputs produced at each stage of our Graph RAG framework, and Figure 5 gives the generated patches along with the gold patch. To evaluate the effectiveness of graph modality in assisting solving practical issues, we also compare the patches generated by CGM with and without code graph (as shown in Figure 5). For the latter, the input of CGM is only the context files provided by Reranker, and does not include the subgraph generated by Retriever.

---

**Issue:**

Use Admin Inline verbose_name as default for Inline verbose_name_plural Description Django allows specification of a verbose_name and a verbose_name_plural for Inline classes in admin views. However, verbose_name_plural for an Inline is not currently based on a specified verbose_name. Instead, it continues to be based on the model name, or an a verbose_name specified in the model's Meta class. This was confusing to me initially (I didn't understand why I had to specify both name forms for an Inline if I wanted to overrule the default name), and seems inconsistent with the approach for a model's Meta class (which does automatically base the plural form on a specified verbose_name). I propose that verbose_name_plural for an Inline class should by default be based on the verbose_name for an Inline if that is specified. I have written a patch to implement this, including tests. Would be happy to submit that.

**Rewriter:**

```
"rewriter_inferer_output": [ "File names containing 'admin' AND classes related
to 'Inline' to locate the definition of Inline classes.", "Functions or methods
handling the setting of 'verbose_name' and 'verbose_name_plural' within 'admin'
directory to find where these attributes are processed.", "Code snippets where
'InlineModelAdmin' is defined or used to identify the base class for admin
inlines.", "File name containing 'options.py' AND functions related to
'get_verbose_name_plural' to find where the pluralization logic is currently
implemented.", "Functions or methods handling the initialization of Inline
classes within 'admin' directory to find where default values are set." ],
"rewriter_extractor_output_entity": [ "django/contrib/admin/options.py" ],
"rewriter_extractor_output_keyword": [ "verbose_name", "verbose_name_plural",
"Inline" ]
```

**Retriever:**

Key Results **django.util.text**

```
{
    "nodeType": "File",
    "id": 52874,
    "fileName": "text.py",
    "filePath": "django/utils",
    "text": "import html.entities\nimport tags and … "
}
```

```
{
    "nodeType": "File",
    "id": 53224,
    "fileName": "__init__.py",
    "filePath": "django/utils/translation",
    "text": "\"\"\"\nInternationalization support … "
}
```

**Reranker:**

```
"FineRanker": {
    "django/contrib/admin/options.py": {
        "system_prompt": "You are an experienced software developer ...",
        "user_prompt": "\n<repository>\ndjango/django\n</repository>...",
        "response": "[start_of_analysis]\nThe issue describes a ..."
    },
```

---

Figure 4: The given issue and the intermediate outputs produced by Rewriter, Retriever, and Reranker, respectively.

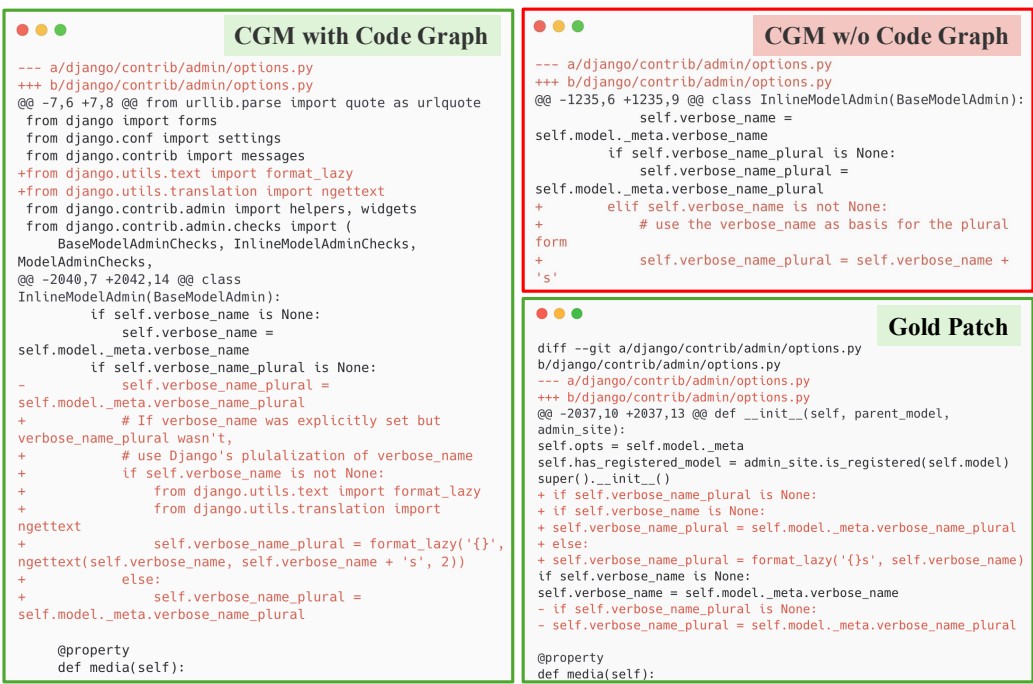

Figure 5: Patches generated by CGM (with or without code graph), along with the gold patch. Green boxes represent successful patches and the red box represents unsuccessful one.

## E   Error Analysis

In issue-fixing task, errors can be classified into two types: **execution errors**, where the generated code cannot be successfully executed, and **unresolved cases**, where the code is executable but does not fix the issue.

Notably, the vast majority ($\approx 80\%$) of CGM's failures are unresolved cases, not execution errors. This highlights that our architecture's high fidelity in generating syntactically correct and executable code, even when the semantic logic for the fix is not perfect.

Then we manually inspect around $20\%$ ($33/171$) of the failure cases (6 from execution errors and 27 from unresolved ones) to profile the failure patterns and identify potential development directions.

The main reasons for execution errors are: (1) **being misled by complicated issue descriptions** ($60\%$) and (2) **occasional mistakes in syntactic generation** ($40\%$). For example, in the instance `django__django-12113`, model directly copies a diff snippet from the lengthy input issue as output. Breaking down complicated issues into clearer instructions for CGM Reader might alleviate this. As for the second reason, we observe occasional missing functionality, such as a missing `return` clause in instance `sympy__sympy-13043`. The appearance of such errors, while infrequent, is a known characteristic of code large language models.

As a more major error, the unresolved ones are mainly caused by: (1) **limited reasoning ability for complex issues** ($55\%$), (2) **knowledge gap** ($26\%$), and (3) **cascading errors from the RAG module** ($19\%$).

We demonstrate the first reason by instance `scikit-learn__scikit-learn-11040`. Here, CGM does locate and fix the user-reported vulnerable class, `NearestNeighbors`. However, the architecturally superior solution, provided by the golden patch, was to fix its parent class, `NeighborsBase`, from which `NearestNeighbors` inherits. This distinction is not trivial. In fact, in the code graph, there exist an edge connecting `NearestNeighbors` with `NeighborsBase`. However, the LLM decoder in CGM fails to leverage the edge to modify `NeighborsBase` instead of `NearestNeighbors`.

For reason (2), knowledge gap is exemplified by LLM's unawareness of the current internal implementation of third-party packages: in the instance `scikit-learn__scikit-learn-10949`, model attempts an unsupported operation on a `NumPy` array, which could be mitigated by including such details directly into the Code Graph.

Finally, the errors caused by RAG means the files needed to be modified have not been retrieved by previous module (Retriever or Reranker), thus leading to the failure of CGM. In other words, improving the Recall of the GraphRAG module can further improve the final performance of CGM.

## F    Limitations

We have limited this work to Python and Java—two popular object-oriented languages—so the current code graph schema is untested on other paradigms. Although these languages cover a large part of real-world issue-fixing scenarios, extending our framework to other paradigms (e.g., multi-paradigm languages like Rust, or functional languages such as Haskell) will require re-examining how code graphs are built to capture paradigm-specific structures.

## G    Prompt Template Example

This section shows the prompt templates used by Rewriter (Figure 6 and Figure 7) and Reranker (Figure 8 and Figure 9) in our Graph RAG framework.

**Prompts:**
<issue>
{ISSUE TEXT}
</issue>
This is an issue related to repository '{REPO NAME}'.

**Instructions:**
1. Analysis:
- Analyze the provided issue description. Identify the relevant File, Class, or Function involved.
- Determine the specific problem or error encountered and note any clues that may assist in locating the relevant or problematic area.
2. Extraction:
- After the analysis, extract ALL the mentioned code entities (File, Class, or Function), especially Files.
- Then extract three potential and meaningful keywords, responding in the following format:

[start_of_analysis]
<detailed_analysis>
[end_of_analysis]

[start_of_related_code_entities]
<entity_name_with_path>
[end_of_related_code_entities]

[start_of_related_keywords]
<keywords>
[end_of_related_keywords]

**Notes:**
- Pay attention to the information in the error logs (if exists).
- The buggy code exists solely in the project described in the issue (e.g., django, sklearn). Buggy location is usually not in the tests files or external packages.
- Your extracted entities should be CONCISE, ACCURATE and INFORMATIVE.
- Provide the relative path for code entities if specified (e.g., package/foo.py). Relative path is relative to the repository itself, do not include suffix like '/home/username/', '/etc/service/' or '/tree/master'.
- Do not include any additional information such as line numbers or explanations in your extraction result.

**Preferred extraction Examples of Code Entities:**
- repo/cart.py
- Class User()
- def getData()

**Preferred extraction Examples of Keywords:**
- train_loop
- hooks
- docker

**Unpreferred extraction Examples of keywords:**
- something wrong
- input validation
- TypeError

Figure 6: Prompt for Extractor in Rewriter.

**Prompts:**
<issue>
{ISSUE TEXT}
</issue>
This is an issue related to repository '{REPO NAME}'.
**Task:**
Based on the issue description provided, identify the characteristics of code entities (files, functions, class) that might need to be modified.
For each characteristic, generate a search query that could help locate relevant code entities in a codebase.
**Instructions:**
First, analyze the issue description and identify keywords, features, and functionalities that are likely relevant to the modification of code entities.
Then, create queries that capture these characteristics, focusing on:
- File names that may implement relevant functionalities.
- Functions or methods that are related to the features described in the issue.
- Any patterns or structures that might be relevant to the functionalities mentioned.
For example:
- File related to the initialization of a neural network.
- Function related to the training process.
- Code used to configure the service.
Please answer in the following format:

[start_of_analysis]
<detailed_analysis>
[end_of_analysis]

[start_of_related_queries]
query 1:
query 2:
...
[end_of_related_queries]

**Notes:**
- Your queries should be DETAILED, ACCURATE and INFORMATIVE.
- Your queries should be a complete sentences and do not include additional explanation.
- The number of queries is up to five, so be focus on the important characteristics.
- Your queries should focus on the repository code itself, rather than other information like commit history.
- Pay attention to the information in the error logs (if exists).

**Preferred Query Examples:**
- Look for references to "tqdm" or "progress_bar" within the training loop files to find where progress bars are currently updated.
- Code snippets where 'gethostbyname' function from 'socket' module is called.
- File name containing 'mysql.py' AND functions related to 'MySQLStatementSamples' initialization.
- Functions or methods handling hostname resolution or encoding within 'datadog_checks' directory.
- Find all occurrences of "early_stopping" within files that also mention "Trainer" to identify where early stopping logic is implemented and potentially needs adjustment for non-default 'val_check_interval'.

Figure 7: Prompt for Inferer in Rewriter.

**Prompts:**
You are an experienced software developer who specializes in extracting the most relevant files for solving issues from many reference files.

**Task:**
Based on the information received about the issue from a repository, find the most likely few files from among those that may be able to resolve the issue.

**Instructions:**
1. Analysis:
- Analyze the provided issue description and files, and pay attention to the relevance of the provided files with the given issue, especially those might be modified during fixing the issue.
- Determine the specific problem or error mentioned in the issue and note any clues that could help your judgment.
2. Extraction:
- Based on your analysis, choose the Top **10** relevant files which might be used in fixing the issue.
- You should choose files from the provided files, and should not modify their name in any way.

Respond in the following format:
[start_of_analysis]
<detailed_analysis>
[end_of_analysis]

[start_of_relevant_files]
1. <file_with_its_path>
2. <file_with_its_path>
3. ...
[end_of_relevant_files]

**Notes:**
- You can refer to to the information in the error logs (if exists).
- The relevant file usually exists in the project described in the issue (e.g., django, sklearn). File need modification is usually not in the tests files or external packages.
- The file you choose should be contained in the provided files.
- Provide the file path with files. Do not include redundant suffix like '/home/username/', '/etc/service/' or '/tree/master'.
- Do not include any additional information such as line numbers or explanations in your extraction result.
- Files for initialization and configuration might be modified during changing the code.

**Preferred extraction Examples of Related Files:**
1. src/utils/file_handler.py
2. core/services/service_manager.py
3. ...

<repository>
{REPO NAME}
</repository>

<issue>
{ISSUE TEXT}
</issue>

<reference_python_file_list>
{REFERENCE PYTHON FILES}
</reference_python_file_list>

<other_reference_file_list>
{OTHER REFERENCE FILES}
</other_reference_file_list>

Figure 8: Prompt for Reranker in Stage 1.

**Prompts:**
You are an experienced software developer who specializes in assessing the relevance of the file for solving the issue in software repositories.

**Task:**
For a file provided, evaluate the likelihood that modifying this file would resolve the given issue, and assign a score based on specific criteria.

**Instructions:**
1. Analysis:
- Analyze the provided issue description and the content of the single relevant file, pay attention to any keywords, error messages, or specific functionalities mentioned that relate to the file.
- Determine how closely the contents and functionality of the file are tied to the problem or error described in the issue.
- Consider the role of the file in the overall project structure (e.g., configuration files, core logic files versus test files, or utility scripts).
2. Scoring:
- Based on your analysis, assign a score from 1 to 5 that represents the relevance of modifying the given file in order to solve the issue.

**Score Specifications:**
1. **Score 1**: The file is almost certainly unrelated to the issue, with no apparent connection to the functionality or error described in the issue.
2. **Score 2**: The file may be tangentially related, but modifying it is unlikely to resolve the issue directly; possible in rare edge cases.
3. **Score 3**: The file has some relevance to the issue; it might interact with the affected functionality indirectly and tweaking it could be part of a broader fix.
4. **Score 4**: The file is likely related to the issue; it includes code that interacts directly with the functionality in question and could plausibly contain bugs that lead to the issue.
5. **Score 5**: The file is very likely the root cause or heavily involved in the issue and modifying it should directly address the error or problem mentioned.

Respond in the following format:
[start_of_analysis]
<detailed_analysis>
[end_of_analysis]

[start_of_score]
Score <number>
[end_of_score]

**Notes:**
- The content of the file shows only the structure of this file, including the names of the classes and functions defined in this file.
- You can refer to to the information in the error logs (if exists).

<repository>
{REPO NAME}
</repository>

<issue>
{ISSUE TEXT}
</issue>

<file_name>
{FILE NAME}
</file_name>

<file_content>
{FILE CONTENT}
</file_content>

Figure 9: Prompt for Reranker in Stage 2.

