# OpenReview forum: "Code Graph Model (CGM): A Graph-Integrated Large Language Model for Repository-Level Software Engineering Tasks"
_NeurIPS.cc/2025/Conference — NeurIPS 2025 poster_

### Official Review · Reviewer_HU7w · 2025-06-10

**Clarity:** 2
**Significance:** 3
**Originality:** 2
**Rating:** 3
**Confidence:** 3

**Summary:**

This paper offers a code graph model (CGM) system for repo-level SWE automation. It addressed a critical gap of open source LLMs in handling repo level tasks.

**Questions:**

Is it possible for the authors to reorganize the paper? I believe the content is quite deep and possibly with immense value. But, the presentation and organization perhaps need improvement.

**Ethical Concerns:**

["NO or VERY MINOR ethics concerns only"]

**Limitations:**

The paper is Appendix-heavy. I think the authors should reconsider the paper organization.

**Quality:**

2

**Strengths And Weaknesses:**

The paper addresses a critical need. The methodology section is particularly strong, especially the Code Graph Construction and Code Graph Model components.

However, I find that the Experiment and Results sections are inadequate and lack clearly defined research questions. Currently, the paper proposes a method and immediately presents Experiment and Results without sufficient context. I recommend authors to address following questions:

1. What is the baseline?
2. Which factors contributed most significantly to the overall success of the proposed approach?
3. What are the fundamental research questions?
4. I would like to see a clear assessment of the differential contribution of the graph-aware attention mask. Specifically, what would the performance be using a vanilla attention mask instead? The same applies to the graph-based retriever. These two artifacts are among the most intriguing elements of the paper.

That said, there is a wealth of valuable information in the Appendix, especially Figures 4 and 5. I recommend the authors carefully review their paper and present it following a clear scientific structure: Hypotheses → Experiments → Results → Conclusion.

---

> ### Author Rebuttal · Authors · 2025-07-30
>
> _W1: what is the baseline_
>
>
>
> There are two scenarios in our experiments - repository-level issue fixing (Section 5.1) and repository-level code completion (Section 5.2).
>
> **1. For Repository-Level Issue Fixing (Section 5.1, Tables 1 & 2):**
>
> Our primary baselines are the **SOTA methods from the official SWE-bench Lite leaderboards (as of May 2025)**. This ensures we compare our work against the most competitive and relevant systems in the field. To provide a fair and nuanced assessment of our open-source, agentless approach, we categorize our comparisons as follows:
>
> + **Against Open-Source Systems:** We compare CGM directly with other fully open-source systems, regardless of the underlying LLM they use. As shown in **Table 1(a)**, our method (CGM-SWE-PY) ranks 2nd overall among all open systems on SWE-bench Lite.
> + **Against Methods Using Open-Weight Models:** This is our most direct comparison group. Here, CGM-SWE-PY ranks **1st**, outperforming the next-best method (Moatless+DeepSeek-V3) by a significant margin of 12.33%.
> + **Against All Systems (Including Proprietary Agents):** We also situate our performance in the broader context of all published results, including those using powerful, closed-source agentic frameworks. Our method achieves a top-8 overall rank, demonstrating its competitiveness even against proprietary systems.
>
> **2. For Repository-Level Code Completion (Section 5.2, Tables 3 & 4):**
>
> Here, we compare against two categories of strong baselines to isolate the benefits of our CGM architecture:
>
> + **SOTA LLMs:** In **Table 3**, we compare CGM against powerful standalone language models (e.g., DeepSeek-236B, Mistral-123B) using a standard retrieval method. This demonstrates that the architectural innovations of CGM provide a substantial performance boost beyond what can be achieved by simply using a powerful LLM backbone.
> + **SOTA RAG Methods:** In **Table 4**, we conduct a head-to-head comparison against established RAG and context-retrieval methods for code, such as BM25, R2C2, and RepoFuse. This is a critical comparison, as it shows that CGM's method of directly integrating graph structure into the model is more effective than methods that also use graphs for retrieval but then flatten the context into linear text.
>
> These baseline selections were deliberately chosen to rigorously test our central hypotheses and clearly demonstrate the unique advantages of the CGM framework. Full details of all baseline implementations are provided in **Appendix C.6.4 and C.6.5**.
>
>
>
>
>
> _W2 & W4 & L1: most significantly contributed factors & the contribution of the graph-aware attention mask and the graph-based retriever & appendix heavy_
>
>
>
> To clarify, we have conducted comprehensive ablation studies in Appendix C.7, listed as follows:
>
> + **Ablation of the high-level Graph RAG Framework**: Verifying the contribution of each pipeline component.
>     - **Rewriter**: Removing it dropped performance by 8.33%, confirming its value in clarifying issue descriptions.
>     - **Retriever**: Its removal caused an 11.33% performance drop, highlighting its role in finding relevant subgraphs.
>     - **Reranker**: Proved the most critical component; its removal led to a 24.67% performance drop.
>     - **Full RAG Pipeline (w/o R3)**: Removing all three modules caused a 33.33% drop, showing the RAG pipeline is essential to filter noise.
> + Ablation of the Core CGM Reader: Analyzing the model's architecture, training, and generalization.
>     - **Model Architecture**:
>         * **Semantic Integration**: Progressively fine-tuning the model's Adapter, Decoder, and Encoder showed that training all components is essential for maximum performance.
>         * **Structural Integration**: Replacing the graph-aware attention mask with the original causal mask hurt performance by 8.61% (Java) and 5.56% (Python).
>         * **Graph Modality**: Replacing the graph modality input with pure text input (by flattening the text-rich graph as in RepoGraph and GraphCoder) resulted in a 37.67% performance decrease, validating CGM's superior design.
>     - **Training Strategy**:
>         * **Pre-training Task**: Removing the subgraph reconstruction task reduced performance by 7.65%, confirming its importance.
>     - **Generalization**:
>         * **Backbone Models**: CGM’s performance scales with the underlying model's coding ability across Llama3.1-70B-Instruct, Qwen2.5-Coder-32B/7B-Instruct, and Qwen2.5-72B, consistently enhancing each backbone.
>
>
>
> From the above results, we can see that **among all high-level graph RAG components, the Reranker has the most significant impact**—removing it results in a 24.67% performance drop. This highlights the importance of precise file selection for effective issue resolution.
>
>
>
> Within the core CGM Reader, as shown in Table 11, **joint fine-tuning of the adapter, encoder, and decoder (i.e., semantic integration) brings the largest performance improvement**. Notably, the adapter’s ability to map encoded node representations to the LLM input space is crucial for maximizing effectiveness.
>
>
>
> Regarding the **graph-aware attention mask (i.e., the ablation study of structural integration)**, using vanilla causal attention mask instead of the graph-aware attention mask leads to performance degradation on both Java and Python dataset (EM performance drops by 8.61% and 5.56%, respectively), as shown in Table 11.
>
>
>
> Omitting the **graph-based Retriever**—by relying solely on the Rewriter and Reranker for file selection—resulted in an 11.33% reduction in issue fixing performance on SWE-bench Lite (see Table 10). This underlines the Retriever’s role in supplying contextually relevant, structurally coherent subgraphs for downstream modules.
>
>
>
> Due to space constraints in the initial version, we were only able to include the most essential ablation findings in Section 5.3 of the main paper. For the camera-ready version (which allows one additional page), **we will move more comprehensive ablation results and analyses into the main content, as you suggested.**
>
>
>
> _W3: fundamental research questions_
>
>
>
> Our research is driven by observing the current landscape of automated software engineering, which is dominated by complex, agent-based systems that rely on proprietary LLMs (as discussed in Lines 26-36). This paradigm, while powerful, introduces significant challenges related to accessibility, cost, data privacy, and the inherent unpredictability of multi-step agentic reasoning.
>
>
>
> This context gives rise to our primary research question, stated explicitly in the Introduction (**Lines 37-38**):
> "**Can open-source LLMs be employed in an agentless manner to effectively complete repository-level coding tasks?**"
>
>
>
>
>
> _W5&Q1: reorganize the paper_
>
>
>
> We agree that a clear presentation following the **Hypothesis → Experiments → Results → Conclusion** framework is essential for conveying our contributions effectively. We believe our paper already adheres to this logical structure, and we appreciate the opportunity to clarify this flow.
>
> Here is how the current sections of our paper map to the scientific structure suggested by the reviewer:
>
> 1. **Hypotheses (Section 1: Introduction):**
> Our primary hypothesis is explicitly stated as a research question in the introduction: **"Can open-source LLMs be employed in an agentless manner to complete repository-level coding tasks?"** (Lines 37-38). We further posit a core technical hypothesis that this is achievable **if the LLM is empowered to comprehend the repository’s structural dependencies via a code graph** (Lines 42-45, "the key lies in empowering the open-source LLMs to fully comprehend code repositories... but also the dependencies across functions and files").
> 2. **Methodology to Test the Hypotheses (Sections 3 & 4):**
> To test our hypotheses, we designed and built the **Code Graph Model (CGM)** and the accompanying **Graph RAG framework**.
>     - **Section 3** details the construction of the code graph, the fundamental data structure required by our hypothesis.
>     - **Section 4** describes the novel CGM architecture and the Graph RAG framework, which are the instruments we use to conduct our experiments. This section explains _how_ we integrate semantic and structural information to test if this approach is effective.
> 3. **Experiments & Results (Section 5):**
> This section is dedicated to the empirical validation of our hypotheses.
>     - We test our CGM on challenging repository-level tasks (Issue Fixing in Sec 5.1, Code Completion in Sec 5.2) and present the outcomes in **Tables 1, 2, 3, and 4**.
>     - Crucially, the **Ablation Studies (Section 5.3 and Appendix C.7)** directly test the core components of our technical hypothesis by measuring the performance impact of our structural integration (the graph-aware attention mask) and other key design choices. The results presented there provide strong evidence supporting our claims.
> 4. **Conclusion (Section 6):**
> This section summarizes our findings, directly answering our initial research question. We conclude that our graph-integrated, agentless approach using open-source models is not only viable but also highly competitive, thus validating our central hypotheses.
>
> While we have not used explicit "Hypothesis" or "Results" subheadings, we believe this logical progression is inherent in the paper's current organization. We hope this clarification helps illustrate the scientific rigor of our paper's structure, and we thank the reviewer again for prompting this helpful discussion.

---

> ### Author Response · Authors · 2025-08-06
> **Your Feedback on Our Paper**
>
> Dear Reviewer HU7w,
>
> Thank you for your positive feedback; we were very encouraged to hear that you are satisfied with our rebuttal!
>
> While we appreciate your satisfaction with our response, we noticed the score seemed unchanged. We would like to know if this suggests there are other aspects of our work that we could still strengthen.
>
> We are looking forward to hear from you.
>
> Best regards,
>
> Authors

---

### Official Review · Reviewer_Fgkk · 2025-06-22

**Clarity:** 2
**Significance:** 3
**Originality:** 4
**Rating:** 5
**Confidence:** 2

**Summary:**

Summary

The paper proposes a code graph-based model to resolve repository-level issue resolution problems like SWE Bench tasks.

1. Their technique is a fixed workflow-based approach and does not depend on the agent to take autonomous steps.
2. They model the entire repository as a graph structure, including the code repository structure, inheritance, and call graph structure.
3. They have multiple steps in their procedure -

They comprehend and extract relevant entities from the natural language issue.

They perform a search on the code graph structure and extract the relevant files that include the nodes in the graph.

They perform re-ranking to pick the top most k files from the retrieved files.

They provide the selected graph structure as input to the LLM and modify the attention matrix supplied to the LLM. They then pass the actual file contents to the LLM. And ask the LLM to generate the final patch file.

**Questions:**

In Table 4, for the purposes of retrieval, were all the components used except for the final Reader? As the CGM reader is trained to output the final answer? If the Reader was also used during retrieval, how is the Reader used for the purposes of retrieval?

**Ethical Concerns:**

["NO or VERY MINOR ethics concerns only"]

**Final Justification:**

I appreciate the authors’ thoughtful and timely rebuttal. It clarified several points from my review and addressed some questions at a high level (like the role of the "Reader"). Balancing the helpful clarifications against the remaining limitations, my overall evaluation is unchanged, and I maintain my original score.

**Limitations:**

Yes, for limitations.

No, for potential societal impacts. The implications of open source models on the issue of automation should be considered in the paper.

**Quality:**

3

**Strengths And Weaknesses:**

The writing of the paper can be improved; it would have been nice to start with a motivating example and to explain the methodology using that example. As the proposed method is very involved and complicated, understanding the paper was challenging.

Strengths

1. The work is significant as it proposes an open-source trained LLM that achieves on-par performance with closed-source models.

2. The paper introduces a novel technique to integrate code graph structures into an LLM; they explicitly train the LLM to understand the code graph structure and to reconstruct the file contents given the code graph structure.

3. They show impressive performance on the SWE Bench Lite benchmark as well as the CrossCodeEval benchmark.

Weakness

1. There are a lot of moving components in the proposed approach. It is not exactly clear what components of the approach can be used for tasks other than issue resolution. Please also see my question on what component is used for retrieval-based generation.

---

> ### Author Rebuttal · Authors · 2025-07-30
>
> _W1 & Q1: Graph RAG for retrieval-based generation._
>
> The core of our contribution is the CGM, which is a powerful graph-aware generator (or "Reader"). The Graph RAG framework (Rewriter, Retriever, Reranker) is a flexible front-end designed to supply the CGM with the most relevant context, and its components can be adapted or simplified depending on the task's complexity.
>
> To clarify how this works in practice for our two experimental tasks:
>
> 1. **For Complex Issue Fixing (Section 5.1):** This task starts with an ambiguous natural language query (an issue report). Therefore, we employ the **full Graph RAG pipeline (Rewriter, Retriever, Reranker)** to progressively refine the query and localize the most relevant subgraph to feed into the CGM. The CGM's role here is purely as the final **Reader/Generator**.
> 2. **For Code Completion (Section 5.2):** This task is simpler as the location for code generation is already known. Consequently, the full, sophisticated RAG pipeline is unnecessary. We instead use a simplified retrieval strategy, as explicitly described in **Section 5.2 (Lines 334-336)** and **Appendix C.6.1**. Specifically, we directly build the input subgraph by retrieving the **one-hop neighbors** of the target function from the code graph. This subgraph, along with the incomplete file, is then fed into the **CGM, which acts solely as the generator.**
>
> To ensure our comparisons in the code completion experiments were fair and informative, we designed them as follows:
>
> 1. **Comparison against other LLMs (Table 3):** To isolate the power of the CGM as a generator, we **fixed the retrieval method for all models**. We used the same one-hop graph retrieval for CGM and all baseline LLMs (Mixtral, DeepSeek, etc.). For the baseline models that cannot process graphs, we flattened the retrieved subgraph into text. The results, which show CGM's superior performance on ComplexCodeEval, demonstrate its enhanced capability as a generator, even when the retrieval strategy is held constant.
> 2. **Comparison against other RAG methods (Table 4):** To assess our end-to-end performance, we compared our simplified pipeline (one-hop graph retrieval + CGM generator) against other specialized RAG baselines (e.g., BM25, R2C2, RepoFuse). As detailed in **Section 5.2 (Lines 342-347)** and **Appendix C.6.4**, each baseline uses its own distinct retrieval mechanism. The results show that our graph-based approach typically outperforms these methods, highlighting the effectiveness of our method.
>
> To directly answer your final question: **The CGM is exclusively used as the final generator in all experiments. It was never used for retrieval.** The components of the Graph RAG framework are responsible for retrieval, and their application is adapted to the specific task. We will ensure this distinction is further clarified in the final version of the paper.
>
> _L1: The implications of open source models on the issue of automation_
>
> Our work is motivated by the significant challenges that the current reliance on proprietary, closed-source models poses to the automation of software engineering. We have explicitly discussed these implications in several key sections:
>
> 1. **Problem Formulation (Introduction, Lines 33-36 & 85-87):** We directly address the limitations of closed-source models, which currently dominate the field. We highlight that their use creates substantial barriers for the research and engineering communities, including:
>     - **Limited Accessibility & High Cost:** Preventing widespread adoption and experimentation.
>     - **Lack of Customization:** Prohibiting researchers and organizations from fine-tuning models on proprietary or domain-specific codebases.
>     - **Data Privacy & Security Risks:** Making it infeasible for many companies to use these models with their sensitive, internal code repositories.
> 2. **Core Research Question (Introduction, Lines 37-41):** Based on these implications, we frame our entire investigation around the central question: **"Can open-source LLMs be employed in an agentless manner to complete repository-level coding tasks?"** This question positions our work as a direct response to the need for more accessible and transparent automation tools.
> 3. **Contribution and Impact (Conclusion, Lines 376-377):** Our conclusion reinforces this theme by stating that our work "establishes a new direction for developing powerful, transparent, and accessible tools for automated SE." By demonstrating that a fully open-source framework can achieve state-of-the-art results, we argue that our approach democratizes access to powerful SE automation, fosters a more transparent and extensible ecosystem, and provides a practical solution for real-world deployment where security and customization are paramount.
>
> In summary, our paper is not merely an application of an open-source model; it is a dedicated effort to demonstrate its viability and to provide the community with a practical, powerful, and transparent pathway for future research in automated software engineering. We will ensure this narrative is further emphasized in the final version.

---

### Official Review · Reviewer_2ymT · 2025-06-29

**Clarity:** 2
**Significance:** 2
**Originality:** 2
**Rating:** 4
**Confidence:** 3

**Summary:**

This paper introduces Code Graph Models (CGMs), which is a graph-enhanced Large Language Model (LLM) architecture, for comprehensive repository-level code understanding. This architecture integrates repository code graph structures into the LLM’s attention mechanism and map node attributes to the LLM’s input space using a specialized adapter. When combined with an agentless graph RAG framework, this approach achieves a 43.00% resolution rate on the SWE-bench Lite benchmark using the open-source Qwen2.5-72B model.

**Questions:**

1. Can you add more in-depth analysis in the ablation study? Also, I would suggest adjusting the main content and adding the results of ablation study to the main content, as they are important.
2. Can you use 2-3 sentences to summarize your strengths that are superior to previous works like RepoGraph and GraphCoder?

**Ethical Concerns:**

["NO or VERY MINOR ethics concerns only"]

**Final Justification:**

The rebuttal addresses my main concerns. Thus I have raised my ratings.

**Limitations:**

1. This work might provide more results using different open-sourced LLMs to demonstrate the effectiveness of the proposed approach.
2. This paper might add a new section to discuss the broader impact of this approach to give more insights.

**Quality:**

2

**Strengths And Weaknesses:**

Strengths:
1. This work proposes an agentless approach to effectively address repository-level code tasks.
2. When combined with an agentless graph RAG framework, this approach achieves a 43.00% resolution rate on the SWE-bench Lite benchmark using the open-source Qwen2.5-72B model, which ranks first among open weight models, second among methods with open-source systems, and eighth overall, surpassing the previous best open-source model-based method by 12.33%.
3. The codebase is released for reproducibility.

Weaknesses:
1. Although there are engineering contributions in this work, the proposed approach seems too complicated to follow. Detailed ablation study and in-depth analysis are required in the main content to help readers gain more insight on the architecture design.
2. This work mentions some related work such as RepoGraph[1] and GraphCoder[2] also leverage a graph-based approach to address repository-level code tasks, however, this work neither conducts a comprehensive performance comparison nor gives sufficient discussions on why the proposed approach is better.
3. The experimental results are too limited to confirm the effectiveness of the proposed approach. We are not clear whether the effectiveness of CGM comes from the open-sourced code LLM Qwen2.5-72B or the architecture itself, since this paper only presents results from Qwen2.5-72B. More code LLMs such as DeepSeek-Coder and CodeLlama can be used for comparisons.
4. Although this work claims the approach outperforms a couple of agentless baselines, there is an obvious gap between this approach and those agent-based baselines. The paper does not offer a compelling justification for why its proposed method should be adopted over agent-based approaches in applications, especially given that the latter demonstrate superior performance.

[1] Ouyang et al., RepoGraph: Enhancing AI Software Engineering with Repository-level Code Graph, ICLR2025.
[2] Liu et al., GraphCoder: Enhancing Repository-Level Code Completion via Coarse-to-fine Retrieval Based on Code Context Graph. ASE 2024.

---

> ### Author Rebuttal · Authors · 2025-07-30
>
> _W1&Q1: only engineering contributions & detailed ablation study_
>
> First of all, we would like to clarify that, beyond engineering contributions, we propose a novel method to integrate the graph modality with the language modality (i.e., LLMs). Specifically,
>
> > (i) **Semantic Integration**: Node attributes (containing code or comments) are first encoded by a pretrained text encoder and then mapped to the LLM's input space via an adapter, enabling the model to understand the semantic information of all nodes.
> >
>
> > (ii) **Structural Integration**: The graph structure is incorporated into the LLM through the attention mask, allowing direct message passing only between neighboring nodes in each layer of the LLM, similar to spatial Graph Neural Networks (GNNs).
> >
>
> Moreover, we have conducted comprehensive ablation studies in Appendix C.7, listed as follows:
>
> + **Ablation of the high-level Graph RAG Framework**: Verifying the contribution of each pipeline component.
>     - **Rewriter**: Removing it dropped performance by 8.33%, confirming its value in clarifying issue descriptions.
>     - **Retriever**: Its removal caused an 11.33% performance drop, highlighting its role in finding relevant subgraphs.
>     - **Reranker**: Proved the most critical component; its removal led to a 24.67% performance drop.
>     - **Full RAG Pipeline (w/o R3)**: Removing all three modules caused a 33.33% drop, showing the RAG pipeline is essential to filter noise.
> + Ablation of the Core CGM Reader: Analyzing the model's architecture, training, and generalization.
>     - **Model Architecture**:
>         * **Semantic Integration**: Progressively fine-tuning the model's Adapter, Decoder, and Encoder showed that training all components is essential for maximum performance.
>         * **Structural Integration**: Replacing the graph-aware attention mask with the original causal mask hurt performance by 8.61% (Java) and 5.56% (Python).
>         * **Graph Modality**: Replacing the graph modality input with pure text input (by flattening the text-rich graph as in RepoGraph and GraphCoder) resulted in a 37.67% performance decrease, validating CGM's superior design.
>     - **Training Strategy**:
>         * **Pre-training Task**: Removing the subgraph reconstruction task reduced performance by 7.65%, confirming its importance.
>     - **Generalization**:
>         * **Backbone Models**: CGM’s performance scales with the underlying model's coding ability across Llama3.1-70B-Instruct, Qwen2.5-Coder-32B/7B-Instruct, and Qwen2.5-72B, consistently enhancing each backbone.
>
> Due to space constraints in the initial version, we were only able to include the most essential ablation findings in Section 5.3 of the main paper. For the camera-ready version (which allows one additional page), **we will move more comprehensive ablation results and analyses into the main content, as you suggested.**
>
> _W2&Q2: discussion and comparison with RepoGraph and GraphCoder_
>
> We would like to clarify that we have discussed the differences between our proposed CGM and previous graph-based methods such as RepoGraph [45] and GraphCoder [49] in Section 2 (Lines 141–147):
>
> > Recent research has also focused on enhancing code understanding by incorporating structural information through graph-enhanced repository modeling [45,48,49]. However, even when graph structures are used during retrieval, existing methods typically **flatten the retrieved code snippets into linear text sequences for downstream model prompting**. This flattening process fails to preserve the inherent heterogeneity between graph and text modalities. As a remedy, we propose the CGM that explicitly aligns these two distinct modalities, enabling better preservation and utilization of structural information throughout the entire process.
> >
>
> To address performance comparison, RepoGraph achieves 29.67% on SWE-bench Lite (using GPT-4o as its base model), as reported in their paper. Our CGM attains 43.00% on the same benchmark using the open-source Qwen2.5-72B, demonstrating a significant improvement. **We will update Table 1 and the corresponding discussion** to include these direct comparisons for greater clarity.
>
> To summarize the strengths of CGM over RepoGraph and GraphCoder in two sentences:
> **CGM enables decoder-only LLMs to explicitly understand and attend over graph structures, going beyond simple textual flattening of code graphs. Additionally, by compressing node text into compact embeddings, CGM can efficiently encode much larger repository graphs than previous approaches, leading to superior performance and scalability.**
>
> Finally, as further evidence, our ablation study (Lines 810–814) shows that flattening the graph into plain text for LLM input leads to a 37.67% performance decrease, underscoring the importance of our explicit graph integration.
>
> _W3 & L1: effectiveness of the achitecture itself & more backbones_
>
> To clarify, Table 1(a) shows that even when using the **same Qwen2.5-72B backbone**, methods such as SWE-Fixer and Lingma SWE-GPT achieve issue resolution rates of only 24.67% and 22.00% respectively on SWE-bench Lite, while our proposed CGM achieves a significantly higher 43.00%. This large margin demonstrates that **the gains stem primarily from our architecture, not just the choice of backbone**.
>
> Furthermore, as discussed in Appendix C.8, we have also trained CGM on other LLM backbones, including Llama3.1-70B-Instruct, Qwen2.5-Coder-32B-Instruct, and Qwen2.5-Coder-7B-Instruct, to test the generalizability of our approach. Notably, CGM improves the performance of Llama3.1-70B-Instruct from 7% (as reported in Lingma SWE-GPT [1]) to 25.33% on SWE-bench Lite—an absolute improvement of 18.33%. These results further confirm that the effectiveness of CGM comes from its architectural innovations, independent of the underlying LLM.
>
> _W4: justification for use against agent-based methods_
>
> We would like to clarify the positioning and advantages of our approach compared to both open-source and closed-source agent-based methods.
>
> For agent-based methods using open-source models, their performance is generally lower than CGM’s. As shown in Table 1(a) on SWE-bench Lite, CGM outperforms the strongest open-source agent-based baseline (Moatless+DeepSeek-V3) by 12.33%, despite DeepSeek-V3 itself being a stronger model than Qwen2.5-72B, which we use as our backbone. This highlights the advantage of our architecture over even the best open-source agent frameworks (see Lines 298–303, Page 8).
>
> For agents based on closed-source model, the reliance on closed-source models creates substantial barriers for the broader SE community [13,14], including limited accessibility, inability to enhance or customize models for specific tasks, and serious security concerns regarding the privacy of sensitive code repositories when interacting with external API services. This point is mentioned in Lines 33-36. A more comprehensive discussion regarding agent-based methods and their practical challenges and limitations is provided in Lines 112-128 (Agent-drive Methods for SE in Section 2).
>
> At the time of submission, our method ranks eighth overall on SWE-bench Lite, making it highly competitive even compared to the latest agent-based approaches. Importantly, CGM delivers these results using a fully open-source, agentless workflow, ensuring transparency, extensibility, and real-world deployability without the constraints of commercial API access. We believe this is meaningful for both AI and SE communities. Furthermore, the CGM framework is inherently flexible and **can be further improved as stronger encoders, adapters, and LLM backbones become available**, which is likely to further narrow (or even close) the gap to the very top results dominated by proprietary agents.
>
> _L2: broader impact_
>
> Following your suggestion, we will add the following discussion to the paper:
>
> > This research introduces a powerful new paradigm for reasoning about complex, interconnected information by deeply integrating text-rich graph structures directly into Large Language Models. Instead of perceiving a software repository as a flat, linear sequence of text, our approach endows the LLM with an explicit, structured understanding of the codebase's architecture. This is crucial for advancing AI in software engineering, as the model can now explicitly reason about the intricate web of dependencies—how functions call each other, classes inherit properties, and files are interconnected. Consequently, the model can better anticipate the downstream impact of a proposed code change, a reasoning ability previously exclusive to experienced human developers. Furthermore, the code graph itself serves as a valuable asset for human oversight, providing a clear visualization that helps developers grasp complex repository structures and validate the model’s logic.
> >
>
> > The implications of this approach, however, extend far beyond software engineering, offering a blueprint for enhancing LLM reasoning in any domain that can be represented as a text-rich graph. In academic research, for instance, a citation network where papers are nodes and citations are edges could be ingested. This would allow a model to not only summarize a paper's content but also understand its intellectual lineage, identify seminal works, and even forecast emerging research trends by analyzing its structural position within the scientific community. Similarly, in e-commerce, a graph of products and user interactions would enable hyper-personalized recommendation engines that understand the nuanced relationships between items based on both their descriptions and their connections in purchasing patterns. By fusing the structural context of a graph with the deep semantic understanding of an LLM, this methodology unlocks a new tier of contextual intelligence applicable across countless fields.
> >

---

> > ### Comment · Reviewer_2ymT · 2025-08-02
> >
> > Thanks to the authors for the rebuttal.
> >
> > I still have some questions and concerns:
> > 1. Thanks for the response to W1. Since the ablation study helps readers figure out why the proposed method works and plays a crucial role in an empirical research paper, I would strongly suggest authors to make a big edit and move this part to the main content, instead of Appendix.
> > 2. The rebuttal mentions that "we propose a novel method to integrate the graph modality with the language modality (i.e., LLMs)" and it looks like an engineering solution. However, I am still confused about what the main research question is in this paper, how this paper design experiments and analyses to investigate this question, and what the conclusions are.

---

> ### Author Response · Authors · 2025-08-03
> **Reply to Reviewer 2ymT (Part 1)**
>
> We thank the reviewer for their instant feedback and the opportunity to provide further clarification.
>
> _Q1: Move ablation study to main content_
>
> We completely agree. The camera-ready version allows an extra page, and we will use this space to expand the description of our ablation study.
>
> _Q2: Our research question_
>
> This is a question that **requires a detailed response**, and we appreciate the opportunity to elaborate on the core research question that drives our work, the principled design of our method, and how our experiments validate our scientific claims.
>
> ###### 1. **The Core Research Question**
> Our paper's central research question is: **"Can open-source LLMs be employed in an agentless manner to effectively complete repository-level coding tasks?"** (Lines 37-38).
>
> The fundamental challenge (Lines 42-45) we address is the mismatch between the sequential nature of LLMs and the complex, non-linear structure of code repositories. While human programmers rely on understanding hierarchical and reference dependencies (e.g., callers, callees, inheritance) to navigate code, LLMs traditionally struggle with this relational information. We posit that enabling an LLM to natively comprehend this structure is key to unlocking its full potential for software engineering.

---

> ### Author Response · Authors · 2025-08-03
> **Reply to Reviewer 2ymT (Part 2)**
>
> ###### 2. **Our Novelty: A Principled, Architectural Solution**
> To investigate this question, we proposed the **Code Graph Model (CGM)**. We argue this is a novel architectural contribution, not just an engineering pipeline. Our approach is conceptually analogous to how **Vision-Language Models (VLMs)** integrate distinct modalities (e.g., images and text) into a unified architecture. In our work, we replace the vision modality with the **graph modality** to represent a code repository, which required us to develop specialized techniques for this unique data type.
>
> + **Why Code Graph? The Need for a Structured Representation.** Before integration, a repository must be properly represented. We use a **code graph** (Section 3) because it explicitly captures the structural information that is critical for complex coding tasks. The graph's edges represent both hierarchical (`contains`) and reference (`calls`, `imports`) dependencies, mirroring how a developer examines a function's context—its outer class, its callers, and its callees—before editing the function itself. The graph nodes, in turn, hold the semantic information (the source code itself).
> + **Why CGM? Bridging the Modality Gap.** Simply creating a graph is insufficient; the LLM must be able to reason over it. Existing methods that "flatten" the graph into text inevitably lose the rich structural heterogeneity and struggle with the context length limitations of LLMs. Our CGM architecture bridges this modality gap through two key innovations:
>     1. **Semantic Integration via Node-Level Compression:** A primary challenge is fitting the rich content of a repository graph into the finite context window of an LLM. If we were to simply feed the raw source code from each graph node into the LLM decoder, the text would be tokenized into a large number of tokens, quickly exhausting the model's context window and limiting its view to only a small fraction of the repository. Our semantic integration module addresses this directly. It encodes the entire source text of a graph node (which can be hundreds of tokens long) into a single, compact **node token** via a specialized encoder and adapter. This compression is a key part of our design, as it **effectively expands the LLM decoder's context length by orders of magnitude**. It allows the model to simultaneously consider the semantic information from a vast number of code entities (nodes), enabling it to reason over a much larger and more complete subgraph than would be possible with raw text.
>     2. **Structural Integration via Graph-Aware Attention:** We integrate the graph's topology directly into the LLM by replacing the standard causal attention mask with a **graph-aware attention mask** derived from the repository's adjacency matrix. This forces the model's attention mechanism to operate along the actual dependencies of the code graph, ensuring that information flows only between neighboring nodes. This mimics the message-passing of a Graph Neural Network (GNN) within the Transformer architecture, enabling genuine structural reasoning. **The benefit of this becomes clear in a practical task.** When CGM is tasked with modifying a specific function, the generation process for the new code attends to the corresponding "node token" for that function. Crucially, due to the graph-aware attention mechanism, this single node token has already aggregated contextual information from its neighbors (e.g., its callers, callees, and parent class) during the LLM's forward pass. This means that the model's prediction for the very next token is implicitly conditioned on the function's entire structural context. In essence, the model **modifies the function with direct reference to its dependencies**, which explains the performance improvements we observe on complex issue-fixing benchmarks like SWE-bench Lite, where understanding such relationships is paramount.
>
> By integrating the graph modality and considering its special properties (like the adjacency matrix) in our model design, we believe our work is a significant step beyond a mere engineering solution.

---

> ### Author Response · Authors · 2025-08-03
> **Reply to Reviewer 2ymT (Part 3)**
>
> ###### 3. **Experimental Design and Validation**
> Our experiments were designed to systematically validate our hypothesis and answer our research question.
>
> + **Primary Evaluation:** We tested CGM on challenging repository-level tasks, including Issue Fixing (Sec 5.1) and Code Completion (Sec 5.2), using standard benchmarks. As shown in Tables 1, 2, 3, and 4, CGM significantly outperforms existing open-source methods, demonstrating that our graph-integrated architecture is highly effective in practice.
> + **Ablation Studies:** To isolate the impact of our novel components, we conducted extensive ablation studies (Section 5.3 and Appendix C.7). These studies dissect the sources of our performance gains and confirm that our core architectural innovations are crucial. For example, we showed that removing the **graph-aware attention mask** or disabling the fine-tuned **semantic integration** module significantly degrades performance. This provides strong evidence that the architectural integration itself, not just the surrounding RAG framework, is the primary driver of our model's success.
>
> ###### 4. **Conclusion**
> Our conclusion in Section 6 directly answers our initial research question. We confirm that by architecturally enhancing an LLM to be graph-aware, it is indeed possible for an open-source, agentless model to serve as a viable and highly competitive solution for repository-level tasks. Our work demonstrates that moving beyond text-only representations and toward true graph-language models is a powerful and promising direction for the future of automated software engineering.
>
>
>
> We hope this detailed explanation clarifies the scientific contributions of our paper. Thank you again for your feedback.

---

> ### Author Response · Authors · 2025-08-06
> **Follow-up on Your Additional Questions**
>
> Dear Reviewer 2ymT,
>
> Thank you again for the valuable discussion! We just wanted to check if it helped to address the further concerns you raised.
>
> We appreciate all your guidance.
>
> Best regards,
>
> Authors

---

### Official Review · Reviewer_7ALZ · 2025-07-04

**Clarity:** 3
**Significance:** 3
**Originality:** 3
**Rating:** 5
**Confidence:** 3

**Summary:**

This paper presents the Code Graph Model (CGM), a framework that integrates code graph structures with open - source Large Language Models (LLMs) to address repository - level software engineering tasks. Traditional approaches relying on proprietary LLM agents face issues like unpredictability and limited accessibility. CGM overcomes these by enabling LLMs to understand codebase functions, files, and their dependencies through semantic and structural integration. It constructs a code graph for each repository, encoding node attributes and incorporating graph structures into the LLM's attention mechanism. Coupled with an agentless Graph Retrieval - Augmented Generation (RAG) framework, CGM achieves a 43.00% resolution rate on the SWE - bench Lite benchmark using the open - source Qwen2.5 - 72B model. This performance ranks highly among open - source models and demonstrates the feasibility of open - source LLMs in handling complex repository - level tasks without agents.

**Questions:**

1. Please submit the results or statistical curves of multiple TTS (e.g., different temperature/top-k) sampling runs, showing the potential improvement and stability of the model’s performance under best-of-n sampling. If CGM demonstrates significant improvement under TTS strategies, this will directly raise my evaluation of the model’s capability and the experimental thoroughness of the paper.
2. If the authors can provide a detailed analysis of failure cases—including failure statistics, attribution of failure patterns, and concrete discussion of possible remedies—it will greatly increase my confidence in the research depth and the prospects for future improvement of this work.
3. The resolution rate of CGM-Multi on Java repositories is much lower than on Python , but the reasons were not thoroughly analyzed. Is this due to insufficient support for Java-specific syntax (such as annotations or generics) in the code graph construction process, or is it because the Java sample proportion in training data is low? If possible, could you supplement experiments with other languages in the multi SWE bench[1]? If the current approach cannot be applied to other languages, please clearly state this limitation in the discussion, and provide a road map if available.
4. Have the authors conducted any analysis of how the amount of training data affects repair performance? If so, please consider presenting this analysis to demonstrate that the capabilities of the CGM approach can continue to improve with increased data.

[1]Multi-SWE-bench: A Multilingual Benchmark for Issue Resolving

**Ethical Concerns:**

["NO or VERY MINOR ethics concerns only"]

**Final Justification:**

Because the authors provided a detailed analysis of TTS performance, a bad‐case analysis, and an explanation of why Java underperforms Python, along with the necessary clarifications, I believe the paper’s validity has been further strengthened and my concerns have been well resolved. Therefore, my final judgment is **5: Accept**.

**Limitations:**

yes

**Quality:**

3

**Strengths And Weaknesses:**

## **Strengths：**

1. **Innovative Architectural Design:** Introduces Code Graph Models (CGM), achieving deep integration of code repository graph structures with open-source LLMs through semantic encoders and structure-aware attention mechanisms, solving the structural information loss problem caused by code graph linearization in traditional methods.

2. **Efficient Agent-free Framework:** The developed Graph RAG framework requires only 4 modules  significantly reducing interaction complexity while achieving a 43% problem-solving rate on SWE-bench Lite using open-source models, surpassing closed-source model approaches like Claude 3.5 Sonnet.

3. **Multimodal Information Integration Advantages:** Extends model context length by 512 times through a "chunk encoding-adapter projection" mechanism that compresses long text nodes into LLM-processable node representations, enabling simultaneous processing of graph structures (global dependencies) and complete code text (local details).

4. **Strong Empirical Support:** Validates effectiveness in Python/Java cross-language scenarios, exceeding similar open-source model methods by 12.33% on SWE-bench Lite, while improving ES metrics by 19.8% in code completion tasks, demonstrating method generalization capability.

5. **Interpretability Improvements:** Maintains stable performance even with 10% noisy inputs through controlled noise fine-tuning and structured attention masking, addressing the error accumulation pain point of traditional agent-based methods.

## Weaknesses

1. **No Analysis of TTS Performance:** Many frameworks (such as Agentless) employ test-time scaling to explore the diversity of model outputs. In the experimental setup of CGM, only a single sample was performed, without multiple sampling to explore the upper bound of the model’s capabilities.
2. **Severe Performance Degradation for Small Models:** When using the Qwen2.5-Coder-7B-Ins model, only a 4% resolution success rate was achieved. In comparison, models of similar size such as Seed-Coder-7B-Ins can achieve a score of about 20%.
3. **Limited Language Paradigms:** Experiments and tool design are conducted only on Python and Java (object-oriented languages). There is no exploration of the feasibility for functional, scripting, or more complex paradigms, and cross-paradigm transfer remains unaddressed.
4. **Long Execution Time:** Constructing a repository-level code graph takes about 3 minutes. Combined with model inference time, this is significantly longer compared to other workflow-based frameworks.
5. **Lack of Bad Case Analysis:** The paper lacks in-depth analysis of failure cases. Understanding why CGM fails in certain situations, such as incorrect handling of complex dependencies or loss of structural information, is crucial for further improvement.

---

> ### Author Rebuttal · Authors · 2025-07-31
>
> _W1&Q1: analysis of TTS performance_
>
> Thanks for your insightful suggestion! We have evaluated CGM's performance from Pass@1 to Pass@3 on both SWE-bench Lite and Verified dataset. Our results below show a clear and positive trend: multiple sampling boosts the resolution rate by approximately 2-3% on both datasets compared to single sampling (Pass@1).
> | **Pass@K** | **SWE-Bench Lite** | **SWE-Bench Verified** |
> | :---: | :---: | :---: |
> | K=1 | 43.00% | 50.40% |
> | K=2 | 44.33% | 51.40% |
> | K=3 | 46.67% | 53.20% |
>
> This confirms CGM's performance scales with multiple samples. We will add this analysis to the final version.
>
> _W2: severe performance drop for small models_
>
> First, we want to clarify that the comparison between our 4% result and the ~20% from Seed-Coder is indirect. Seed-Coder's 19.2% result [R1] was achieved on the SWE-bench **Verified** dataset, which is known to yield higher scores than the more challenging SWE-bench **Lite** dataset used for our Table 12 ablation.
>
> To create a direct and meaningful comparison, we established two crucial baselines for the **Qwen2.5-Coder-7B** model in an agentless setting:
>
> 1. Performance on **Verified**: The Seed-Coder report itself shows that the base **Qwen2.5-Coder-7B** model, when used with the agentless workflow from [2], scores only **4.2%** on SWE-bench **Verified**.
> 2. Performance on **Lite**: We further replicate this exact agentless workflow [2] with the **Qwen2.5-Coder-7B** model and tested it directly on SWE-bench **Lite**, achieving a resolution rate of **3.67%**.
>
> In this direct, like-for-like comparison on SWE-bench **Lite**, our **CGM with the same backbone achieves 4.0%**, showing a slight performance improvement over this SOTA agentless baseline.
>
> These results strongly suggest the **primary limitation is the intrinsic capability of the Qwen2.5-Coder-7B model itself**, limiting the performance across all frameworks built upon it. Moreover, our findings highlight that the **CGM architecture benefits from stronger LLM decoders**. As shown in Table 12, scaling the backbone from Qwen2.5-Coder-7B to Qwen2.5-72B-Instruct substantially increases CGM’s resolution rate from 4.0% to 43.0%.
>
> _W3: limited language paradigms_
>
> We agree that expanding our graph-based framework to additional paradigms—such as functional, scripting, or other complex programming styles—is an important direction for future research.
>
> Our focus on Python and Java was determined primarily by the availability of high-quality benchmarks with testing environments. At the time of our experiments, the main multilingual benchmark initiative was an early release of what has since become **Multi-SWE-bench** [2]. **Before its formal release in April 2025, only the Java component (SWE-bench-java Verified [1]) was accessible for comprehensive evaluation, which is why it served as the basis for our Java experiments.** The full Multi-SWE-bench, with expanded language and paradigm coverage, was released too close to the submission deadline to allow for the extensive data collection and model training required for robust experimentation.
>
> **We have acknowledged this object-oriented focus as a limitation in Appendix E** and will expand this discussion in the final version. We are eager to extend CGM to cover a broader range of languages and paradigms by using the new Multi-SWE-bench in our future work.
>
> _W4: long execution time_
>
> We would like to clarify that our current graph construction tool is a prototype, and there are several clear avenues for optimization. For example, we can parallelize the code parsing phase—followed by a final, serialized step for node relation establishment—to significantly accelerate graph generation. Moreover, after initial construction, the graph can be maintained through lightweight, incremental updates: when code is modified, affected nodes and edges are removed, and only the changed code is re-parsed and reinserted.
>
> With these engineering improvements, we anticipate reducing graph generation time to about 10 seconds for small projects (under 100K lines of code) and 30 seconds for large projects (over 100K LOC).
>
> According to SWE-Gym [44], agent-based methods require at least 20 interaction turns for comparable performance, whereas our approach uses fewer than 5 LLM calls. Therefore, its overall runtime is substantially shorter, even accounting for the optimized graph construction and intermediate processing.
>
> **We will add this point in Appendix B.**
>
> _W5&Q2: Bad case analysis_
>
> Thanks for your insightful input! Here we provide the **evaluation report** (generated by the official SWE-bench toolkit) for CGM-SWE-PY on SWE-bench Lite, followed by an **error analysis** to diagnose CGM's limitations..
>
> **Evaluation Report**
>
> The toolkit shows CGM achieves an overall 43.00% (129/300) resolution rate. The detailed results by **repository** (e.g., performance on django or sympy) and **year** (year of issue origin) are below.
>
> | **Repository** | **Resolved Rate** |
> | :---: | :---: |
> | astropy | 1/6 |
> | django | 62/114 |
> | matplotlib | 8/23 |
> | seaborn | 2/4 |
> | flask | 0/3 |
> | requests | 3/6 |
> | xarray | 1/5 |
> | pylint | 3/6 |
> | pytest | 6/17 |
> | scikit-learn | 12/23 |
> | sphinx | 3/16 |
> | sympy | 28/77 |
>
>
> | **Year** | **Resolved Rate** |
> | :---: | :---: |
> | 2012 | 1/1 |
> | 2013 | 1/3 |
> | 2014 | 0/1 |
> | 2015 | 1/4 |
> | 2016 | 5/16 |
> | 2017 | 6/21 |
> | 2018 | 26/59 |
> | 2019 | 28/66 |
> | 2020 | 22/42 |
> | 2021 | 26/57 |
> | 2022 | 13/30 |
> | 2023 | 1/1 |
>
>
> CGM's performance is stable over time, indicating no data contamination, while its varied success across repositories suggests the potential influence of project-specific characteristics.
>
> **Error Analysis**
>
> Firstly, the vast majority (~80%) of CGM's failures are **unresolved cases** (i.e., executable but incorrect), not **execution errors**, highlighting its **high fidelity in generating syntactically valid code**, even when the semantic logic for the fix is not perfect.
>
> Then we manually inspect 33 failure cases (20% of total, 6 from execution errors and 27 from unresolved ones) to profile the failure patterns.
>
> 1. The main reasons for **execution errors** are **(1) being misled by complicated issue** (60%) and **(2) occasional mistakes in syntactic generation** (40%). For example, in the _'django__django-12113'_, model directly copies a diff snippet from the issue description; breaking down issue into clearer instructions could alleviate this. As for the second, we observe occasional missing functionality like a missing return clause in _'sympy__sympy-13043'_. The appearance of such errors, while infrequent, is a known characteristic of code LLMs [R3].
> 2. As a more major error, **the unresolved ones** are mainly caused by: **(1) limited reasoning ability for complex issues** (55%), **(2) knowledge gap** (26%), and **(3) cascading errors from the graph RAG module** (19%).
>     1. **Limited Reasoning:** This is demonstrated in _'scikit-learn__scikit-learn-11040'_, where CGM does locate and fix the user-reported vulnerable class, NearestNeighbors. However, the architecturally superior solution, provided by the golden patch, was to fix its parent class, _**KNeighborsMixin**_, from which _NearestNeighbors_ inherits. This distinction is not trivial. In fact, **in the code graph, there exists an edge connecting _NearestNeighbors_ with _KNeighborsMixin_**. Despite it, the current LLM decoder in CGM fails to leverage the edge to modify _KNeighborsMixin_ instead of _NearestNeighbors._
>     2. **Knowledge gap:** This is exemplified by **LLM's unawareness of the current implementation of third-party packages**. In _'scikit-learn__scikit-learn-10949'_, model attempts an unsupported operation on a NumPy array, which could be mitigated by including such details directly in the code graph.
>     3. **Cascading errors**: These errors occur when necessary files have not been retrieved by the previous RAG module, thus leading to the failure of CGM. **Improving the Recall of the GraphRAG module** can further improve CGM's final performance.
>
> This analysis will be added to the final paper.
>
> _Q3: java is worse than python_
>
> While the performance of CGM-Multi is lower on the Java benchmark compared to the Python benchmark, we believe this reflects **the benchmark's higher inherent difficulty**, rather than a specific weakness of our model in processing Java.
>
> This difficulty gap is not unique to our method; it is a consistent trend across other SOTA models in the field. To illustrate this, we can look at the performance of a powerful baseline from the official leaderboards:
>
> + On the **Python-centric SWE-bench Lite**, **SWE-Agent + GPT-4o** achieves a resolution rate of **32.00%** (as seen in Table 1a).
> + On the **SWE-bench-java Verified** benchmark, the **same** **SWE-Agent + GPT-4o** system's performance drops sharply to just **6.59%** (as seen in Table 2).
>
> This demonstrates that the Java benchmark poses a substantially greater challenge for current methods. In this difficult context, our **CGM-Multi achieves a 14.29%** resolution rate, **more than doubling the performance** of the SWE-Agent+GPT-4o baseline. Therefore, while lower than our Python results, our Java performance is SOTA and significantly outperforms other leading methods on that specific, more difficult benchmark.
>
> _Q4: performance vs data amount_
>
> Thanks for pointing this out! To analyze the influence of training data volume, we have trained CGM on subsets of the training data. The results below show a clear positive trend:
>
> | **Training Data** | **SWE-Bench Lite** |
> | :---: | :---: |
> | 10% | 29.67% |
> | 30% | 36.67% |
> | 50% | 39.33% |
> | 100% | 43.00% |
>
> This demonstrates CGM's potential to improve further as more issue-fix pairs are available.
>
> [R1] Seed-Coder: Let the Code Model Curate Data for Itself
>
> [R2] Multi-SWE-bench: A Multilingual Benchmark for Issue Resolving
>
> [R3] LLM Hallucinations in Practical Code Generation: Phenomena, Mechanism, and Mitigation

---

> > ### Comment · Reviewer_7ALZ · 2025-08-05
> > **Thank you for your detailed responses!**
> >
> > Thank you for your detailed responses to the concerns I raised. Thanks to your explanations, I have gained a deeper understanding of CGM’s TTS performance improvements, the failure‐case analysis, and the reasons for Java’s lower performance compared to Python. I hope you will incorporate these improvements into the paper, as they will greatly enhance its validity. Consequently, I have increased my scores.

---

> ### Author Response · Authors · 2025-08-05
>
> Dear Reviewer 7ALZ,
>
> Thank you again for your valuable feedback. We have submitted our rebuttal and hope it addresses your comments.
>
> With the discussion period ending soon, we just wanted to send a gentle reminder. We'd be grateful for a chance to hear your thoughts.
>
> Best regards,
>
> Authors

---

> ### Author Response · Authors · 2025-08-06
> **Thank you!**
>
> Thank you for your positive feedback! We will incorporate these parts into the revised paper.

---

### Note · Authors · 2025-08-13

Dear Reviewers and Area Chair,

We would like to take this opportunity to sincerely thank all reviewers for their valuable time and insightful comments. The discussion phase was highly constructive and has allowed us to further improve the quality of our manuscript.

Inspired by your feedback, we have **enriched the paper with several targeted experiments**—including new analysis on test-time scaling, a detailed failure-case analysis, an investigation into the impact of training data size, and refining several key arguments in the original paper. We believe these additions, particularly those suggested by Reviewer 7ALZ, **have made our work much more robust and complete**.

Last but not least, we would like to briefly highlight our core contribution again:
- **A Novel Method**: We introduce a novel architecture that explicitly aligns graph and code modalities for complex repository-level tasks. By integrating graph topology directly into the LLM's attention mechanism, our model performs deeper structural reasoning, moving beyond text-flattening approaches.
- **SOTA Performance Among Open-source Models**: Our open-source model achieves leading results on the challenging SWE-bench benchmark; meanwhile, it outperforms most agent-based methods and demonstrates significant practical value.

We are confident that the revised manuscript is now a much stronger paper. Thank you once again for your dedication and invaluable feedback throughout this process.

Sincerely,

Authors of Submission #1424

---

### Decision · Program_Chairs · 2025-09-17

**Decision:**

Accept (poster)

**Comment:**

This work argues that LLMs can more effectively process large, structured data when this structure is exposed to them. To establish this, it integrates the graph structure of software repositories into an LLM's inputs via a custom adapter. This setup helps it achieve strong performance on SWE-Bench with a large open-source model.

Reviewers commended the effectiveness of the approach and novelty of the technique. At the same time, they expressed concerns around the writing being hard to follow, which complicated the methodology.

In the rebuttal phase, most of these concerns were addressed through extensive clarifications from the authors. Given this, we recommend this paper for acceptance. We look forward to seeing the textual improvements incorporated in the final version.